# A Decentralized Parallel Algorithm for Training Generative Adversarial Nets

**Mingrui Liu**[†], **Wei Zhang**[‡], **Youssef Mroueh**[‡], **Xiaodong Cui**[‡], **Jerret Ross**[‡], **Tianbao Yang**[†], **Payel Das**[‡]

[†] Department of Computer Science, The University of Iowa, Iowa City, IA, 52242
[‡] IBM T. J. Watson Research Center, Yorktown Heights, NY, 10598, USA
`mingruiliu.ml@gmail.com`

## Abstract

Generative Adversarial Networks (GANs) are a powerful class of generative models in the deep learning community. Current practice on large-scale GAN training utilizes large models and distributed large-batch training strategies, and is implemented on deep learning frameworks (e.g., TensorFlow, PyTorch, etc.) designed in a centralized manner. In the centralized network topology, every worker needs to either directly communicate with the central node or indirectly communicate with all other workers in every iteration. However, when the network bandwidth is low or network latency is high, the performance would be significantly degraded. Despite recent progress on decentralized algorithms for training deep neural networks, it remains unclear whether it is possible to train GANs in a decentralized manner. The main difficulty lies at handling the nonconvex-nonconcave min-max optimization and the decentralized communication simultaneously. In this paper, we address this difficulty by designing the **first gradient-based decentralized parallel algorithm** which allows workers to have multiple rounds of communications in one iteration and to update the discriminator and generator simultaneously, and this design makes it amenable for the convergence analysis of the proposed decentralized algorithm. Theoretically, our proposed decentralized algorithm is able to solve a class of non-convex non-concave min-max problems with provable non-asymptotic convergence to first-order stationary point. Experimental results on GANs demonstrate the effectiveness of the proposed algorithm.

## 1 Introduction

Generative Adversarial Networks (GANs) [1] are very effective at modeling high dimensional data, such as images, but are known to be notoriously difficult to train. Recent research on large-scale GAN training by [2] suggests that distributed large-batch training techniques can be beneficial on large models. Their algorithm is based on a centralized network topology [3, 4], in which each worker computes a local stochastic gradient based on its local data and then sends its gradient to a central node. The central node aggregates the local stochastic gradients together, updates its model parameters by first-order methods and then sends the parameters back to each worker. The central node is the busiest node since it needs to communicate with each worker concurrently. This communication is the main bottleneck of centralized algorithms since it could lead to a communication traffic jam when the network bandwidth is low or network latency is high. To address this issue, decentralized algorithms are usually considered as a surrogate when the cost of centralized communication is prohibitively expensive. In decentralized algorithms, instead, each worker only communicates with its neighbors and a central node is not needed. To this end, recently a decentralized algorithm was also designed for training a deep neural network [5]. In addition, decentralized algorithms only require workers to communicate with their trusted neighbors and are usually a good way for maintaining privacy [6, 7, 8].

While decentralized algorithms are beneficial, they are limited from an optimization perspective. All previous decentralized works are designed either for solving convex and non-convex minimization problems [6, 7, 8, 5] or convex-concave min-max problems [9, 10, 11]. However, none of them are directly applicable for non-convex non-concave min-max problems such as GANs. In this paper, we

design the first gradient-based decentralized algorithm for solving a class of non-convex non-concave min-max problems with non-asymptotic theoretical convergence guarantees, which we verify with numerical experimentation.

Our problem of interest is to solve the following stochastic optimization problem:

$$\min_{\mathbf{u}} \max_{\mathbf{v}} F(\mathbf{u}, \mathbf{v}) := \mathbb{E}_{\xi \sim \mathcal{D}} \left[ f(\mathbf{u}, \mathbf{v}; \xi) \right], \tag{1}$$

where $F(\mathbf{u}, \mathbf{v})$ is possibly non-convex in $\mathbf{u}$ and non-concave in $\mathbf{v}$ while $\xi$ is a random variable following an unknown distribution $\mathcal{D}$. In the context of GANs, $\mathbf{u}$ and $\mathbf{v}$ represent the parameters for the generator and the discriminator respectively. Several works [12, 13, 14, 15] have established a non-asymptotic convergence to an $\epsilon$-first-order stationary point (i.e., a point $(\mathbf{u}, \mathbf{v})$ such that $\|\mathbf{g}(\mathbf{u}, \mathbf{v})\| \leq \epsilon$, where $\mathbf{g}(\mathbf{u}, \mathbf{v}) = [\nabla_{\mathbf{u}} F(\mathbf{u}, \mathbf{v}), -\nabla_{\mathbf{v}} F(\mathbf{u}, \mathbf{v})]^{\top}$) for a class of nonconvex-nonconcave min-max problems under various assumptions. Other works [16, 17, 18, 19] focus on GAN training and good empirical performance. However, all of them are built upon the single-machine setting. Although the naive centralized parallel algorithm in these works can also apply, it suffers from a high communication cost on the busiest node (e.g., parameter server) and has privacy vulnerabilities. Furthermore, it is nontrivial to design a decentralized parallel algorithm for nonconvex-nonconcave min-max problems. This difficulty is due to decentralized communication only being able to achieve partial consensus among workers, which makes analysis of nonconvex-nonconcave min-max optimization difficult. Our contributions are the following:

- We design a decentralized parallel algorithm called Decentralized Parallel Optimistic Stochastic Gradient (DPOSG) for a class of nonconvex-nonconcave min-max problems, in which both primal and dual variables are updated simultaneously using only first-order information. Our main novelty lies in the design of simultaneous update combined with multiple rounds of decentralized communication, which is the key for our theoretical analysis. This particular design also allows us to utilize the random mixing strategy as proposed in [20, 21] to further improve the performance, which is verified by our experiments.

- Under the similar assumptions in [12, 15], we analyze DPOSG and establish its non-asymptotic convergence to $\epsilon$-first-order stationary point. In addition, our algorithm is communication-efficient since the communication complexity on the busiest node is $O\left(\log(1/\epsilon)\right)$. Although our algorithm is designed for a much more complex nonconvex-nonconcave min-max problem, it has only negligible logarithmic communication complexity when compared to the decentralized algorithm for solving nonconvex minimization problems in [5].

- We empirically demonstrate the effectiveness of the proposed algorithm using a variant of DPOSG implementing Adam updates and show a speedup compared with the single machine baseline for different neural network architectures on several benchmark datasets, including WGAN-GP on CIFAR10 [22] and Self-Attention GAN on ImageNet [23].

## 2   Related Work

**Min-max Optimization and GAN Training**   Min-max optimization in convex-concave setting was studied thoroughly by a series of seminal works, including the stochastic mirror descent [24], extragradient method [25, 26], dual extrapolation method [27] and stochastic extragradient method [28].

Recently, a wave of studies for min-max optimization without the convexity-concavity assumption has emerged including nonconvex-concave optimization [29, 30, 31, 32] and nonconvex-nonconcave optimization [33, 12, 13, 14, 15]. In addition, there is a line of work attempting to analyze the behavior of min-max optimization algorithms and their applications in training GANs [34, 35, 36, 37, 38, 16, 39, 17, 18, 40, 41, 19, 42, 43]. However, all of these works focus on the single machine setting. Although it is easy to extend some of the works to a centralized parallel version, none of them can be applied in a decentralized setting.

**Decentralized Optimization**   Decentralized optimization algorithms were first studied in [44, 45, 46, 20], where the information is exchanged along the edges in a communication graph. Decentralized algorithms are usually employed to handle the possible failure of the centralized algorithms and to maintain privacy [6, 8]. Several deterministic algorithms are analyzed in the decentralized manner including decentralized gradient descent [47, 48, 8, 49], decentralized dual averaging [50, 51], Alternating Direction Methods of Multipliers [52, 53, 54, 55, 56, 57], decentralized accelerated

coordinate descent [58], and the exact first-order algorithm [59]. Recently several seminal works have been released [60, 61] providing optimal deterministic decentralized first-order algorithms for convex problems.

In large-scale distributed machine learning, people are usually interested in using stochastic gradient methods to update the model parameters. There is a plethora of work trying to analyze decentralized parallel stochastic gradient methods for convex [62, 63, 64, 65, 66] and nonconvex objectives [67, 5, 68, 69, 70, 71, 72, 73, 74, 75]. In particular, Lian et al [5] is the first paper showing that decentralized parallel stochastic gradient is able to outperform its centralized version for nonconvex smooth problems. Besides decentralized communication, several works further consider other techniques to make the decentralized communication more efficient, including allowing asynchrony [68], compression techniques [69, 73, 66], skipping communication rounds [75], and event-triggered communication [76]. For strongly convex objective with finite-sum structure, several decentralized algorithms using variance reduction techniques have also been proposed [77, 78, 79].

However, all of these works are analyzed for minimization problems and none of them can be applied for the class of nonconvex-nonconcave min-max problems as considered in our paper.

**Decentralized Optimization for Min-max Problems**  There are several works considering decentralized min-max optimization where inner maximum function is taken over a set of agents [80] or the objective function is convex-concave [9, 10, 11].

When we were preparing our manuscript, we became aware of a simultaneous and independent work [81] in which another decentralized algorithm for solving a class of nonconvex-nonconcave min-max problems was proposed. In their work, the algorithm uses implicit updates based on the proximal point method [82] and was shown to converge to stationary point and consensus. However, their algorithm is not gradient-based and requires that the sub-problem induced by the proximal point step has a closed-form solution, which is computationally expensive and may not hold in practice. It is unclear whether the analysis in [81] can still work if the sub-problem cannot be solved exactly. In contrast, our algorithm's update rule is simple and does not involve any complicated sub-problem solvers, since it only requires to compute a stochastic gradient and then updates the model parameters in each iteration.

## 3   Preliminaries and Notations

We use $\|\cdot\|$ to denote the vector $\ell_2$ norm or the matrix spectral norm depending on the argument. Define $\mathbf{x} = (\mathbf{u}, \mathbf{v})$, $\mathbf{g}(\mathbf{x}) = [\nabla_{\mathbf{u}} F(\mathbf{u}, \mathbf{v}), -\nabla_{\mathbf{v}} F(\mathbf{u}, \mathbf{v})]^\top$. We say $\mathbf{x}$ is $\epsilon$-first-order stationary point if $\|\mathbf{g}(\mathbf{x})\| \leq \epsilon$. At every point $\mathbf{x}$, we only have access to a noisy observation of $\mathbf{g}$, i.e., $\mathbf{g}(\mathbf{x}; \xi) = [\nabla_{\mathbf{u}} f(\mathbf{u}, \mathbf{v}; \xi), -\nabla_{\mathbf{v}} f(\mathbf{u}, \mathbf{v}; \xi)]^\top$, where $\xi$ is a random variable. In the rest of this paper, we use the term *stochastic gradient* and *gradient* to stand for $\mathbf{g}(\mathbf{x}; \xi)$ and $\mathbf{g}(\mathbf{x})$ respectively.

Throughout the paper, we make the following assumption:

**Assumption 1.**   *(i). $\mathbf{g}$ is $L$-Lipschitz continuous, i.e. $\|\mathbf{g}(\mathbf{x}_1) - \mathbf{g}(\mathbf{x}_2)\| \leq L\|\mathbf{x}_1 - \mathbf{x}_2\|$ for $\forall \mathbf{x}_1, \mathbf{x}_2$.*

   *(ii). For $\forall \mathbf{x}$, $\mathbb{E}[\mathbf{g}(\mathbf{x}; \xi)] = \mathbf{g}(\mathbf{x})$, $\mathbb{E}\|\mathbf{g}(\mathbf{x}; \xi) - \mathbf{g}(\mathbf{x})\|^2 \leq \sigma^2$.*

   *(iii). $\|\mathbf{g}(\mathbf{x})\| \leq G$ for $\forall \mathbf{x}$.*

   *(iv). There exists $\mathbf{x}_*$ such that $\langle \mathbf{g}(\mathbf{x}), \mathbf{x} - \mathbf{x}_* \rangle \geq 0$.*

**Remark:** The Assumptions (i), (ii), (iii) are usually made in optimization literature and are standard. The Assumption (iv) is usually used in previous works for solving non-monotone variational inequalities [12] and GAN training [18, 15]. In addition, this assumption holds in some nonconvex minimization problems. For example, it has been shown that this assumption holds in both theory and practice when using SGD for learning neural networks [83, 84, 85].

**Single Machine Algorithm**  Our decentralized algorithm is based on a specific single machine algorithm called Optimistic Stochastic Gradient (OSG) [16, 15] which is designed to solve a class of nonconvex-nonconcave min-max problems. A similar version for convex minimization problem is proposed in [86, 87]. This algorithm keeps two update sequences $\mathbf{z}_k$ and $\mathbf{x}_k$ with the following update rules:

$$\mathbf{z}_k = \mathbf{x}_{k-1} - \eta \mathbf{g}(\mathbf{z}_{k-1}; \xi_{k-1})$$
$$\mathbf{x}_k = \mathbf{x}_{k-1} - \eta \mathbf{g}(\mathbf{z}_k; \xi_k)$$

(2)

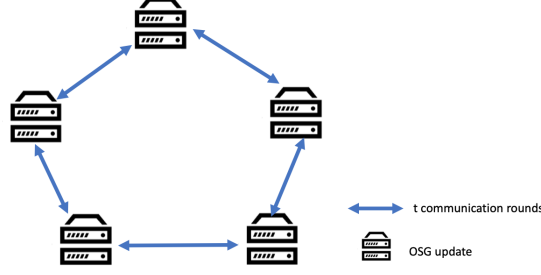

t communication rounds

OSG update

Figure 1: Illustration of DPOSG. Each machine calculates its stochastic gradients and has $t$ communication rounds with its neighbors in parallel. After that each machine conducts OSG update.

It is easy to see that this update is equivalent to the following one line update as in [16]:

$$\mathbf{z}_{k+1} = \mathbf{z}_k - 2\eta \mathbf{g}(\mathbf{z}_k; \xi_k) + \eta \mathbf{g}(\mathbf{z}_{k-1}; \xi_{k-1})$$

## 4    Decentralized Parallel Optimistic Stochastic Gradient

In this section, inspired by the algorithm in [12, 16, 15] in the single-machine setting, we propose an algorithm named Decentralized Parallel Optimistic Stochastic Gradient (DPOSG), which only allows decentralized communications between workers and there is no central node as in the centralized setting which requires communication with each node concurrently in each iteration. Instead, information is only exchanged between neighborhood nodes in the decentralized setting.

Suppose we have $M$ machines. Denote $W \in \mathbb{R}^{M \times M}$ by a doubly stochastic matrix which satisfies $0 \le W_{ij} \le 1, W^\top = W, \sum_{j=1}^{M} W_{ij} = 1$ for $i, j = 1, \ldots, M$. In distributed optimization literature, $W$ is used to characterize the decentralized communication topology, in which $W_{ij}$ characterizes the degree of how node $j$ is able to affect node $i$, and $W_{ij} = 0$ means node $i$ and $j$ are disconnected.

Denote $\lambda_i(\cdot)$ by the $i$-th largest eigenvalue of $W$, then we know that $\lambda_1(W) = 1$. In addition, we assume that $\max\left(|\lambda_2(W)|, |\lambda_M(W)|\right) < 1$.

Denote $\mathbf{z}_k^i \in \mathbb{R}^{d \times 1}$ (and $\mathbf{x}_k^i \in \mathbb{R}^{d \times 1}$) by the parameters in $i$-th machine at $k$-th iteration, and both $\mathbf{z}_k^i$ and $\mathbf{x}_k^i$ have the same shape of trainable parameter of neural networks (the trainable parameters of the discriminator and generator are concatenated together in the GAN setting). Define $Z_k = \left[\mathbf{z}_k^1, \ldots, \mathbf{z}_k^M\right] \in \mathbb{R}^{d \times M}, X_k = \left[\mathbf{x}_k^1, \ldots, \mathbf{x}_k^M\right] \in \mathbb{R}^{d \times M}, \mathbf{g}(Z_k) = \left[\mathbf{g}(\mathbf{z}_k^1), \ldots, \mathbf{g}(\mathbf{z}_k^M)\right] \in \mathbb{R}^{d \times M}$, $\widehat{\mathbf{g}}(\xi_k, Z_k) = \left[\mathbf{g}(\mathbf{z}_k^1; \xi_k^1), \ldots, \mathbf{g}(\mathbf{z}_k^M; \xi_k^M)\right] \in \mathbb{R}^{d \times M}$, where $Z_k, X_k$ are concatenations of all local variables, $\widehat{\mathbf{g}}(Z_k), \mathbf{g}(Z_k)$ are concatenations of all local unbiased stochastic gradients and their corresponding expectations. The Algorithm is presented Algorithm 1, in which every local worker repeatedly executes the following steps (we use machine $i$ at $k$-th iteration as an illustrative example):

- **Sampling:** Sample a minibatch according to $\xi_k^i = \left(\xi_k^{i,1}, \xi_k^{i,2}, \ldots, \xi_k^{i,m}\right)$, where $m$ is the minibatch size.

- **Stochastic Gradient Calculation:** Utilize the sampled data to compute the stochastic gradients for both discriminator and generator respectively, which are $\mathbf{g_u} = \frac{1}{m}\sum_{j=1}^{m} \nabla_{\mathbf{u}} f(\mathbf{u}_k, \mathbf{v}_k, \xi_k^{i,j}), \mathbf{g_v} = \frac{1}{m}\sum_{j=1}^{m} \nabla_{\mathbf{v}} f(\mathbf{u}_k, \mathbf{v}_k, \xi_k^{i,j})$ respectively. Define $\mathbf{g}(\mathbf{z}_k^i; \xi_k^i) = [\mathbf{g_u}, -\mathbf{g_v}]$.

- **Local Averaging and Parameter Update:** Update the model in local memory by $\mathbf{x}_k^i = \tilde{\mathbf{x}}_{k-1}^i - \eta \mathbf{g}(\mathbf{z}_{k-1}^i; \xi_{k-1}^i), \mathbf{z}_k^i = \tilde{\mathbf{x}}_{k-1}^i - \eta \mathbf{g}(\mathbf{z}_k^i; \xi_{k-1}^i)$, where $\tilde{\mathbf{x}}_{k-1}^i$ is calculated via locally averaging the model at $(k-1)$-th iteration over all of its neighbor workers. This local averaging step is done $t$ times according to the matrix $W$. An illustration of this step is in Figure 1.

We make the following remarks on Algorithm 1:

- In both line 3 and line 4, $X_{k-1}W^t$ is the weight averaging step, which can be implemented in parallel with the stochastic gradient calculation step (evaluating $\widehat{\mathbf{g}}(\xi_{k-1}, Z_{k-1})$ and $\widehat{\mathbf{g}}(\xi_k, Z_k)$). When we encounter a large batch in training deep neural networks, the running

---
**Algorithm 1** Decentralized Parallel Optimistic Stochastic Gradient (DPOSG)
---
1: **Input:** $Z_0 = X_0 = \mathbf{0}_{d \times M}$
2: **for** $k = 1, \ldots, N$ **do**
3: $\quad Z_k = X_{k-1}W^t - \eta \cdot \widehat{\mathbf{g}}(\xi_{k-1}, Z_{k-1})$
4: $\quad X_k = X_{k-1}W^t - \eta \cdot \widehat{\mathbf{g}}(\xi_k, Z_k)$
5: **end for**
---

time spent on stochastic gradient calculation usually dominates compared to the weight averaging step during every iteration, so the elapsed time in this case is almost the same as the time spent on the gradient calculation step. This feature makes our algorithm practical and numerically attractive.

- The main differences between the decentralized algorithms for nonconvex-nonconcave min-max problems and minimization problems (e.g., [5]) are two fold. First, we need to introduce two update sequences given in line 3 and line 4 in Algorithm 1, while the algorithm in [5] only requires one update sequence. Second, we need to do the local model averaging $t$ times in each iteration, while one averaging step is sufficient in [5]. Incorporating these ingredients in designing a decentralized algorithm for nonconvex-nonconcave min-max problem is crucial for provable convergence to a stationary point. In addition, we would like to mention that the additional cost and the implementation difficulty incurred by our design are almost negligible compared with [5]. First, the stochastic gradient calculated in line 3 can be reused in the next iteration, which reduces the cost per iteration and shares the similar spirit of one-call stochastic gradient method in [17, 15]. Second, $t$ is only a logarithmic factor of the target accuracy $\epsilon$ to ensure the convergence as shown in Theorem 1 presented later, which possesses almost the same communication cost as in the case of communicating once. Third, our algorithm updates the discriminator and generator simultaneously, and this particular design makes it suitable to implement in the decentralized distributed system as in [5].

- When $W \in \mathbb{R}^{M \times M}$ is a matrix whose every entry is $1/M$ and $t = 1$, our Algorithm 1 recovers the Centralized Parallel Optimistic Stochastic Gradient (CPOSG). The same analysis in [15] can be applied in our case and results in $O(\epsilon^{-4})$ computational complexity and $O(\epsilon^{-2})$ communication complexity on the busiest node.

- When $W \in \mathbb{R}^{M \times M}$ is an identity matrix and $M = 1$, Algorithm 1 recovers the single machine version of Optimistic Stochastic Gradient, which is the same as in (2).

**Theorem 1.** *Suppose Assumption 1 holds and assume* $\|\mathbf{x}_*\| \leq \frac{D}{2}$, $\|\bar{\mathbf{z}}_k\| \leq \frac{D}{2}$ *hold with some* $D > 0$. *Denote* $m$ *by the size of minibatch used in each machine to estimate the stochastic gradient. Define* $\bar{\mathbf{z}}_k = \frac{1}{M}\sum_{i=1}^{M}\mathbf{z}_k^i$ *and* $\rho = \max(|\lambda_2(W)|, |\lambda_M(W)|) < 1$, *where* $\lambda_i(\cdot)$ *stands for the $i$-th largest eigenvalue. Run Algorithm 1 for $N$ iterations, in which* $t \geq \log_{\frac{1}{\rho}}\left(1 + \frac{M\sqrt{mMG^2 + \sigma^2}}{4\sigma}\right)$. *Take* $\eta \leq \min\left(\frac{1}{6\sqrt{2}L}, \frac{1-\rho^t}{\sqrt{32cML}}, \frac{\sqrt{1-\rho^{2t}}}{4m^{1/4}M^{3/4}L}\right)$ *with* $c = 321$. *Then we have*

$$\frac{1}{N}\sum_{k=0}^{N-1}\mathbb{E}\|\mathbf{g}(\bar{\mathbf{z}}_k)\|^2 \leq 8\left(\frac{\|\mathbf{x}_0 - \mathbf{x}_*\|^2}{\eta^2 N} + \frac{20\sigma^2}{mM} + \frac{48(DL\sigma + \sigma^2)}{\sqrt{mM}}\right).$$

Note that our goal is to make sure that $\frac{1}{N}\sum_{k=0}^{N-1}\mathbb{E}\|\mathbf{g}(\bar{\mathbf{z}}_k)\|^2 \leq \epsilon^2$ and Theorem 1 establishes such a non-asymptotic ergodic convergence. In the single-machine setting, to find the $\epsilon$-stationary point, both the stochastic extra-gradient algorithm in [12] and the OSG algorithm in [15] require the minibatch size to be dependent on $\epsilon$. However, in practice, it is not reasonable to assume $m$ to be dependent on $\epsilon$ in single-machine setting since the machine has a memory limit. Handling such a large minibatch could incur some significant system overhead. When $m$ is a constant, independent of $\epsilon$, both the stochastic extragradient algorithm in [12] and the OSG algorithm in [15] cannot be guaranteed to converge to $\epsilon$-stationary point. In the multiple-machines setting, $mM$ is the effective batch size, and we can choose $m$ to be constant and $M$ to be dependent on $\epsilon$ ($M$ can be very large, i.e., our algorithm can handle large number of machines). This insight can be summarized in Corollary 1. In addition, the boundedness assumption of $\|\mathbf{x}_*\|$ and $\|\bar{\mathbf{z}}_k\|$ is a realistic assumption in GAN training.

For example, this assumption explicitly holds due to weight clipping in WGAN [88] training, or implicitly holds by using L2 regularization [89].

**Corollary 1.** *Take* $m = O(1)$, $M = O(\epsilon^{-4})$, $N = O(\epsilon^{-8})$ *in Theorem 1. To find an $\epsilon$-first-order stationary point, Algorithm 1 has $O(\epsilon^{-12})$ computational complexity and $O(t \times$ degree of the network) $= O(\log(1/\epsilon))$ communication complexity on the busiest node.*

**Remark 1** (Non-asymptotic Convergence). *Corollary 1 shows that DPOSG converges to $\epsilon$-stationary point in polynomial time and also enjoys logarithmic communication complexity on the busiest node. The consensus over all nodes can also be achieved due to the logarithmic communication rounds in each iteration.*

**Remark 2** (Spectral Gap and Random Mixing Startegy). *The spectral gap $\rho$ depends on the decentralized communication characterized by matrix $W$. If we use fixed topology where each machine communicates with two neighbors equally, then according to [21], $\rho = \frac{1}{3} + \frac{2}{3}\cos(\frac{2\pi}{M})$ when $M \geq 4$. When using random mixing strategy as in [21], every machine communicates with two random machines each time, and then $\rho$ is much smaller. It implies that $t$ can be chosen to be smaller according to Theorem 1. It further indicates that the Algorithm 1 requires less number of communication rounds in each iteration.*

*We also want to mention that the logarithmic communication complexity does not hold for general ring communication topology, but it indeeds holds for the compelete graph case as studied in [21]. For a fixed ring topology, $\rho$ is close to 1 when $M$ is large, and in this case the per-iteration communication complexity is no longer logarithmic. However, we want to emphasize that it is indeed logarithmic in $\epsilon$ when using the random mixing strategy with a complete graph as in Rand-DP-OAdam in our experiment, in which any two nodes are connected and each node randomly selects two neighbors to communicate $t$ times in each iteration. In this case, it is shown in [21] that $\mathbb{E} \left\| W_1 \dots W_t - \frac{\mathbf{1}_M \mathbf{1}_M^\top}{M} \right\|_2 \leq \frac{\sqrt{M-1}}{(\sqrt{3})^t}$, where $W_i$ represents the communication topology at the $i$-th time, and $\mathbf{1}_M$ stands for a $M$-dimensional vector with all entries being 1. To ensure that RHS$= \frac{\sqrt{M-1}}{(\sqrt{3})^t} \leq \epsilon$, we only need $t = O(\log(1/\epsilon))$ when $M = poly(1/\epsilon)$. Using the random mixing strategy is compatible with the fixed topology as in the proof of Theorem 1, since the two sources of randomness (gradient noise, random mixing) can be decoupled.*

## 4.1 Sketch of the Proof of Theorem 1

We present a high level description here and the detailed proof can be found in Appendix A.4. The key idea in our proof is to approximate the dynamics of a decentralized update to the centralized counterpart, which is of vital importance to establish the convergence of our algorithm. Lemma 1, 2 and 3 are introduced to show how far the decentralized algorithm is away from the centralized counterpart, with proofs included in Appendix A.3. With these lemmas and more refined analysis, we can establish the non-asymptotic convergence of our algorithm. Before introducing those lemmas we introduce the following notations:

**Notations** Define $\epsilon_k^i = \mathbf{g}(\mathbf{z}_k^i; \xi_k^i) - \mathbf{g}(\mathbf{z}_k^i)$, $\widehat{\mathbf{g}}(\epsilon_k^i, \mathbf{z}_k^i) = \mathbf{g}(\mathbf{z}_k^i; \xi_k^1)$, $\widehat{\mathbf{g}}(\epsilon_k, Z_k) = \left[ \mathbf{g}(\mathbf{z}_k^1; \xi_k^1), \dots, \mathbf{g}(\mathbf{z}_k^M; \xi_k^M) \right] \in \mathbb{R}^{d \times M}$, $\bar{\mathbf{z}}_k = \frac{1}{M} \sum_{i=1}^M \mathbf{z}_k^i$. Define $\mathbf{e}_i = [0, \dots, 1, \dots, 0]^\top$, the $i$-th Canonical basis vector. Denote $\mathbf{1}_M$ by a vector of length $M$ whose every entry is 1.

Lemma 1 bounds the squared error between the average of individual gradients on each machine and the gradient evaluated at the averaged weight.

**Lemma 1.** *By taking $\eta = \min \left( \frac{1 - \rho^t}{\sqrt{32cML}}, \frac{\sqrt{1 - \rho^{2t}}}{4m^{1/4}M^{3/4}L} \right)$ with $c \geq 2$, we have*

$$\frac{1}{N} \sum_{k=0}^{N-1} \mathbb{E} \left\| \frac{1}{M} \mathbf{g}(Z_k) \mathbf{1}_M - \mathbf{g}(\bar{\mathbf{z}}_k) \right\|^2 \leq \frac{\sigma^2}{\sqrt{mM}} + \frac{1}{c-1} \cdot \frac{1}{N} \sum_{k=0}^{N-1} \mathbb{E} \left\| \mathbf{g}(\bar{\mathbf{z}}_k) \right\|^2 .$$

The purpose of Lemma 2 is to establish an upper bound for the averaged expected $\ell_2$ error between the averaged stochastic gradient and the individual stochastic gradient on each machine.

**Lemma 2.** *The following inequality holds for the stochastic gradient:*

$$\frac{1}{M} \sum_{i=1}^M \mathbb{E} \left[ \left\| \frac{1}{M} \widehat{\mathbf{g}}(\epsilon_{k-1}, Z_{k-1}) \mathbf{1}_M - \widehat{\mathbf{g}}(\epsilon_{k-1}, Z_{k-1}) \mathbf{e}_i \right\| \right] \leq \frac{2\sigma}{\sqrt{mM}} + 2\mathbb{E} \left[ \frac{1}{M} \sum_{i=1}^M \left\| \mathbf{g}(\mathbf{z}_{k-1}^i) - \mathbf{g}(\bar{\mathbf{z}}_{k-1}) \right\| \right] .$$

Finally Lemma 3 bounds the averaged $\ell_2$ error between the individual gradient on each machine and the gradient evaluated at the averaged weight.

**Lemma 3.** *Define* $\mu_k = \frac{1}{M}\sum_{i=1}^{M}\left\|\mathbf{g}(\mathbf{z}_k^i) - \mathbf{g}(\bar{\mathbf{z}}_k)\right\|$. *Suppose* $\eta < \frac{1}{4L}$ *and* $t \geq \log_{\frac{1}{\rho}}\left(1 + \frac{M\sqrt{mMG^2+\sigma^2}}{4\sigma}\right)$. *We have* $\frac{1}{N}\sum_{k=0}^{N-1}\mathbb{E}\left[\mu_k\right] < \frac{8\eta L\sigma}{\sqrt{mM}(1-4\eta L)}$.

Note that the $\ell_2$ errors in Lemma 2 and Lemma 3 do not appear in the analysis of decentralized algorithms for minimization problems [5]. However, for the nonconvex-nonconcave min-max problem we consider, we need to carefully bound this $\ell_2$ error, which requires $t$ rounds of local decentralized communication.

## 5 Experiments

### 5.1 Experimental Settings

Although our convergence is proved for DPOSG which is not an adaptive algorithm, we implement an adaptive gradient variant of DPOSG implementing Adam updates and its decentralized versions in our experiments, since they provide better empirical performance [16, 15]. We implemented three algorithms: Centralized Parallel Optimistic Adam (CP-OAdam), Decentralized Parallel Optimistic Adam (DP-OAdam) and Randomization Decentralized Parallel Adam (Rand-DP-OAdam) inspired by [20, 21]. We use the term 'learner' to represent 'GPU' in our experiment.

In CP-OAdam, after each minibatch update, every learner sums their weights and calculates the average, which is used as the new set of weights. The summation is implemented using an all-reduce call [1].

We implemented the DP-OAdam algorithm similar to [5]: We arrange all the learners in a communication ring. After each mini-batch update, each learner sends its weights to its left neighbor and right neighbor and sets the average of its weight and the weights of its neighbors as its new weight. In addition, we overlap the gradients computation with the weights exchanging and averaging to further improve run-time performance. An implicit barrier is enforced at the end of each iteration so that every learner advances in a lock-step. Finally, one noteworthy implementation detail is that we arrange the weights update steps for both generator and discriminator such that they occur immediately together. In this way, we could treat the union of two networks as one entire network and plug it in the system that deals with the single objective function as originally proposed in [5].

In Rand-DP-OAdam, we follow the same implementation as in DP-OAdam except that in each iteration each learner randomly selects two neighbors to communicate its weights instead of using a fixed communication topology as in DP-OAdam.

We consider two experiments. The first one is WGAN-GP [22] on CIFAR10 dataset, and the second one is Self-Attention GAN [23] on ImageNet dataset. In each worker, we store both the discriminator and the generator, in which these two neural networks are simultaneously updated in our algorithm. The size of the combined generator and discriminator for WGAN-GP with CIFAR10 is 8.36MB, while the model for Self-Attention GAN with ImageNet is 315.07MB. For both experiments, we tune the learning rate in $\{1\times 10^{-3}, 4\times 10^{-4}, 2\times 10^{-4}, 1\times 10^{-4}, 4\times 10^{-5}, 2\times 10^{-5}, 1\times 10^{-5}\}$ and choose the one which delivers the best performance for the centralized baseline (CP-OAdam), and decentralized algorithms (DP-OAdam, Rand-DP-OAdam) are using the same learning rate as CP-OAdam. For Self-Attention GAN on ImageNet, we tune different learning rates for discriminator and generator respectively and choose to use $10^{-3}$ for generator and $4\times 10^{-5}$ for the discriminator. We fix the total batch size as 256 (i.e. the product of batch size per learner and number of learners is 256).

### 5.2 Convergence and Speedup results in HPC environment

In this section, we conduct experiments in a HPC environment where the network has low latency (1 $\mu$s). We compare convergence and speedup results of DP-OAdam and Rand-DP-OAdam with the centralized baseline CP-OAdam, which is presented in Figure 2. In the two figures on top, the x-axis is number of epochs and the y-axis is the Inception Score (IS-score). In the two figures on bottom, the x-axis is the number of learners and the y-axis is speedup. Since the models on each learner are different at any time, the orange band shows the lowest IS across learners and the highest

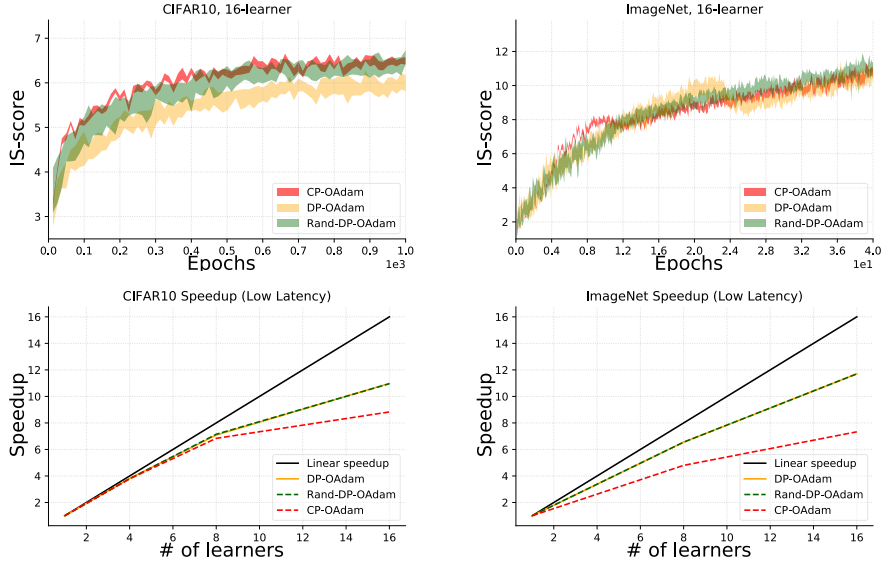

Figure 2: Convergence and speedup comparison between CP-OAdam, DP-OAdam and Rand-DP-OAdam for 16 learners on CIFAR10 and ImageNet. In terms of epochs, DP-OAdam matches the CP-OAdam convergence, and Rand-DP-OAdam further improves over DP-OAdam. Both DP-OAdam and Rand-DP-OAdam have better speedup than CP-OAdam.

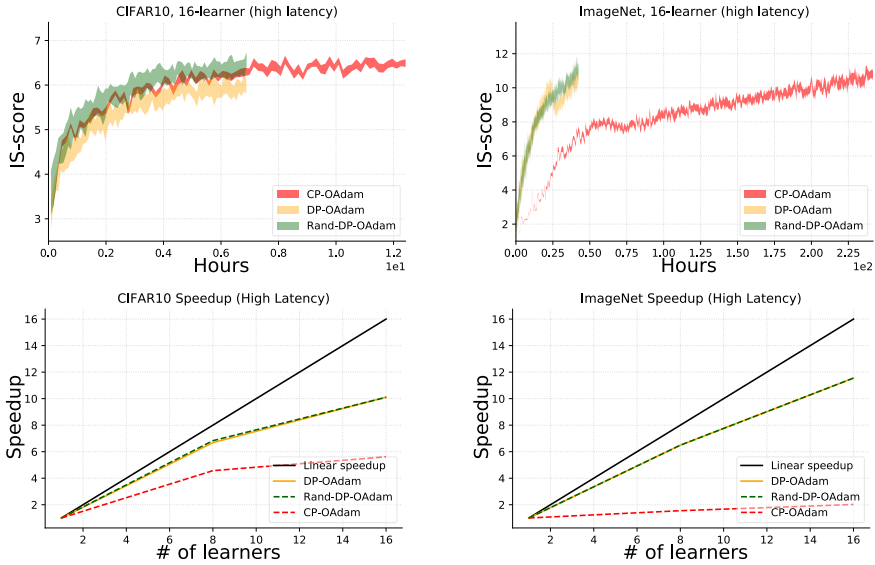

Figure 3: Run-time and speedup comparison for CIFAR10 and ImageNet in a high latency environment. Both DP-OAdam and Rand-DP-OAdam significantly outperform CP-OAdam in terms of both run-time and speedup.

IS across learners when measured for DP-OAdam. Similarly, the green (red) band shows the IS range for Rand-DP-OAdam (CP-OAdam) respectively.

As we can see from Figure 2, in terms of epochs, DP-OAdam for decentralized Parallel GAN training have similar convergence speed as its centralized counterpart (CP-OAdam). Rand-DP-OAdam further improves DP-OAdam significantly due to the usage of random mixing strategy. In terms of speedup, decentralized algorithms (DP-OAdam and Rand-DP-OAdam) are much faster than its centralized counterpart (CP-OAdam). Detailed experiment settings and more experimental results can be found in Appendix B. We also include the generated images in Appendix C.

### 5.3 Run-time and Speedup Results in Cloud Environment

Low-latency network is common in HPC environment whereas high-latency network is common in commodity cloud systems. In this section, we report run-time and speedup results of CIFAR10 and ImageNet experiments on a cloud computing environment where the network has high-latency ($1ms$). The results are presented in Figure 3. Decentralized algorithms (DP-OAdam, Rand-DP-OAdam) deliver significantly better run-time and speedup than CP-OAdam in the high-latency network setting. Allreduce relies on chunking a large message to smaller pieces to enable software pipe-lining to achieve optimal throughput [90], which inevitably leads to many more hand-shake messages on the network than decentralized algorithms and worse performance when latency is high [5, 68, 70]. We recommend practitioners to consider DP-OAdam when deploying Training as a Service system [91] on cloud.

## 6 Conclusion

In this paper, we have introduced a decentralized parallel algorithm for training GANs. Theoretically, our decentralized algorithm is proven to have non-asymptotic convergence guarantees for a class of nonconvex-nonconcave min-max problems. Empirically, our proposed decentralized algorithms are shown to significantly outperform its centralized counterpart and deliver good empirical performance on GAN training tasks on CIFAR10 and ImageNet.

## Acknowledgment

We thank anonymous reviewers for their constructive comments. This work was partially supported by NSF #1933212, NSF CAREER Award #1844403.

## Broader Impact

In this paper, researchers introduce a decentralized parallel algorithm for training Generative Adversarial Nets (GANs). The proposed scheme can be proved to have a non-asymptotic convergence to first-order stationary points in theory, and outperforms centralized counterpart in practice.

Our proposed decentralized algorithm is a class of foundational research, since the algorithm design and analysis are proposed for a general class of nonconvex-nonconcave min-max problems and not necessarily restricted for training GANs. Both the algorithm design and the proof techniques are novel, and it may inspire future research along this direction.

Our decentralized algorithm has broader impacts in a variety of machine learning tasks beyond GAN training. For example, our algorithm is promising in other machine learning problems whose objective function has a min-max structure, such as adversarial training [92], robust machine learning [93], etc.

Our decentralized algorithm can be applied in several real-world applications such as image-to-image generation [94], text-to-image generation [95], face aging [96], photo inpainting [97], dialogue systems [98], etc. In all these applications, GAN training is an indispensable backbone. Training GANs in these applications usually requires to leverage centralized large batch distributed training which could suffer from inefficiency in terms of run-time, and our algorithm is able to address this issue by drastically reducing the running time in the whole training process.

These real-world applications have a broad societal implications. First, it can greatly help people's daily life. For example, many companies provide online service, where an AI chatbot is usually utilized to answer customer's questions. However, the existing chatbot may not be able to fully understand customer's question and its response is usually not good enough. One can adopt our decentralized algorithms to efficiently train a generative adversarial network based on the human-to-human chatting history, and the learned model is expected to answer customer's questions in a better manner. This system can help customers and significantly enhance users' satisfaction. Second, it can help protect users' privacy. One benefit of decentralized algorithms is that it does not need the central node to collect all users' information and every node only communicates with its trusted neighbors. In this case, our proposed decentralized algorithms naturally preserve users' privacy.

We encourage researchers to further investigate the merits and shortcomings of our proposed approach. In particular, we recommend researchers to design new algorithms for training GANs with faster convergence guarantees.

## Footnotes

[1] all-reduce is a reduction operation followed by a broadcast operation. An reduction operation is both associative and commutative (e.g., summation).

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
