[Supplementary Material]

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

 = [0, \ldots, 1, \ldots, 0]^\top$, where 1 appears at the $i$-th coordinate and the rest entries are all zeros. Denote $\mathbf{1}_M$ by a vector of length $M$ whose every entry is 1.

## A.1 Propositions

We first present several propositions which are useful for further analysis.

**Proposition 1** (Lemma 4 in [5]). *For any doubly stochastic matrix $W$ where $0 \leq W_{ij} \leq 1$, $W^\top = W$, $\sum_{j=1}^M W_{ij} = 1$ for $i, j = 1, \ldots, M$. Define $\rho = \max\left(|\lambda_2(W)|, |\lambda_M(W)|\right) < 1$. Then $\left\|\frac{1}{M}\mathbf{1}_M - W^t \mathbf{e}_i\right\| \leq \rho^t$, for $\forall i \in \{1, \ldots, M\}$, $t \in \mathbb{N}$.*

**Proposition 2** (Lemma 5 in [5]).

$$\mathbb{E}\left\|\mathbf{g}(Z_j)\right\|^2 \leq \sum_{h=1}^M 3\mathbb{E}L^2 \left\|\frac{\sum_{i'=1}^M \mathbf{z}_j^i}{M} - \mathbf{z}_j^h\right\|^2 + 3\mathbb{E}\left\|\mathbf{g}\left(\frac{Z_j \mathbf{1}_M}{M}\right)\mathbf{1}_M^\top\right\|^2.$$

## A.2 Useful Lemmas

Inspired by [5], we introduce the Lemma 4 which is useful for our analysis.

**Lemma 4.**

$$\frac{1}{N}\sum_{k=0}^{N-1} \mathbb{E}\left\|\frac{1}{M}\mathbf{g}(Z_k)\mathbf{1}_M - \mathbf{g}(\bar{\mathbf{z}}_k)\right\|^2 \leq \frac{8\eta^2 M L^2 \sigma^2}{(1 - \rho^{2t})\left(1 - \frac{32\eta^2 M L^2}{(1-\rho^t)^2}\right)} + \frac{32\eta^2 M L^2}{(1 - \rho^t)^2\left(1 - \frac{32\eta^2 M L^2}{(1-\rho^t)^2}\right)} \cdot \frac{1}{N}\sum_{k=0}^{N-1} \mathbb{E}\left\|\mathbf{g}(\bar{\mathbf{z}}_k)\right\|^2.$$

*Proof.*

$$\mathbb{E}\left\|\frac{1}{M}\mathbf{g}(Z_k)\mathbf{1}_M - \mathbf{g}(\bar{\mathbf{z}}_k)\right\|^2 \le \frac{1}{M}\sum_{i=1}^{M}\mathbb{E}\left\|\mathbf{g}(\mathbf{z}_k^i) - \mathbf{g}(\bar{\mathbf{z}}_k)\right\|^2 \le \frac{L^2}{M}\sum_{i=1}^{M}\mathbb{E}\left\|\mathbf{z}_k^i - \bar{\mathbf{z}}_k\right\|^2$$

$$= \frac{L^2}{M}\sum_{i=1}^{M}\mathbb{E}\left\|\frac{1}{M}X_{k-1}W^t\mathbf{1}_M - \frac{\eta}{M}\widehat{\mathbf{g}}(\epsilon_{k-1}, Z_{k-1})\mathbf{1}_M - \left(X_{k-1}W^t\mathbf{e}_i - \eta\widehat{\mathbf{g}}(\epsilon_{k-1}, \mathbf{z}_{k-1})\mathbf{e}_i\right)\right\|^2$$

$$\stackrel{(a)}{=} \frac{L^2}{M}\sum_{i=1}^{M}\mathbb{E}\left\|\frac{1}{M}X_0\mathbf{1}_M - \frac{\eta}{M}\sum_{j=1}^{k-1}\widehat{\mathbf{g}}(\epsilon_j, Z_j)\mathbf{1}_M - \frac{\eta}{M}\widehat{\mathbf{g}}(\epsilon_{k-1}, Z_{k-1})\mathbf{1}_M\right.$$

$$\left. - \left(X_0W^{tk}\mathbf{e}_i - \eta\sum_{j=1}^{k-1}\widehat{\mathbf{g}}(\epsilon_j, Z_j)W^{t(k-j)}\mathbf{e}_i - \eta\widehat{\mathbf{g}}(\epsilon_{k-1}, Z_{k-1})\mathbf{e}_i\right)\right\|^2$$

$$\stackrel{(b)}{=} \frac{L^2}{M}\sum_{i=1}^{M}\mathbb{E}\left\|\eta\sum_{j=1}^{k-1}\widehat{\mathbf{g}}(\epsilon_j, Z_j)\left(\frac{\mathbf{1}_M}{M} - W^{t(k-j)}\mathbf{e}_i\right) + \eta\widehat{\mathbf{g}}(\epsilon_{k-1}, Z_{k-1})\left(\frac{\mathbf{1}_M}{M} - \mathbf{e}_i\right)\right\|^2$$

$$\le \frac{2L^2\eta^2}{M}\sum_{i=1}^{M}\mathbb{E}\left\|\sum_{j=1}^{k-1}(\widehat{\mathbf{g}}(\epsilon_j, Z_j) - \mathbf{g}(Z_j))\left(\frac{\mathbf{1}_M}{M} - W^{t(k-j)}\mathbf{e}_i\right) + (\widehat{\mathbf{g}}(\epsilon_{k-1}, Z_{k-1}) - \mathbf{g}(Z_{k-1}))\left(\frac{\mathbf{1}_M}{M} - \mathbf{e}_i\right)\right\|^2$$

$$+ \frac{2L^2\eta^2}{M}\sum_{i=1}^{M}\mathbb{E}\left\|\sum_{j=1}^{k-1}\mathbf{g}(Z_j)\left(\frac{\mathbf{1}_M}{M} - W^{t(k-j)}\mathbf{e}_i\right) + \left(\mathbf{g}(Z_{k-1})\left(\frac{\mathbf{1}_M}{M} - \mathbf{e}_i\right)\right)\right\|^2$$

$$\tag{3}$$

where (a) holds since $W\mathbf{1}_M = \mathbf{1}_M$, (b) holds since $X_0 = \mathbf{0}_{d\times M}$. Note that

$$\mathbb{E}\left\|\sum_{j=1}^{k-1}(\widehat{\mathbf{g}}(\epsilon_j, Z_j) - \mathbf{g}(Z_j))\left(\frac{\mathbf{1}_M}{M} - W^{t(k-j)}\mathbf{e}_i\right) + (\widehat{\mathbf{g}}(\epsilon_{k-1}, Z_{k-1}) - \mathbf{g}(Z_{k-1}))\left(\frac{\mathbf{1}_M}{M} - \mathbf{e}_i\right)\right\|^2$$

$$\stackrel{(a)}{=} \sum_{j=1}^{k-1}\mathbb{E}\left\|(\widehat{\mathbf{g}}(\epsilon_j, Z_j) - \mathbf{g}(Z_j))\left(\frac{\mathbf{1}_M}{M} - W^{t(k-j)}\mathbf{e}_i\right)\right\|^2 + \mathbb{E}\left\|(\widehat{\mathbf{g}}(\epsilon_{k-1}, Z_{k-1}) - \mathbf{g}(Z_{k-1}))\left(\frac{\mathbf{1}_M}{M} - \mathbf{e}_i\right)\right\|^2$$

$$+ 2\mathbb{E}\left\langle(\widehat{\mathbf{g}}(\epsilon_{k-1}, Z_{k-1}) - \mathbf{g}(Z_{k-1}))\left(\frac{\mathbf{1}_M}{M} - W^t\mathbf{e}_i\right), (\widehat{\mathbf{g}}(\epsilon_{k-1}, Z_{k-1}) - \mathbf{g}(Z_{k-1}))\left(\frac{\mathbf{1}_M}{M} - \mathbf{e}_i\right)\right\rangle$$

$$\le \sum_{j=1}^{k-1}\mathbb{E}\left\|(\widehat{\mathbf{g}}(\epsilon_j, Z_j) - \mathbf{g}(Z_j))\right\|^2 \cdot \left\|\left(\frac{\mathbf{1}_M}{M} - W^{t(k-j)}\mathbf{e}_i\right)\right\|^2 + \mathbb{E}\left\|(\widehat{\mathbf{g}}(\epsilon_{k-1}, Z_{k-1}) - \mathbf{g}(Z_{k-1}))\right\|^2 \cdot \left\|\left(\frac{\mathbf{1}_M}{M} - \mathbf{e}_i\right)\right\|^2$$

$$+ 2\mathbb{E}\|\widehat{\mathbf{g}}(\epsilon_{k-1}, Z_{k-1}) - \mathbf{g}(Z_{k-1})\|^2 \cdot \left\|\frac{\mathbf{1}_M}{M} - W^t\mathbf{e}_i\right\| \cdot \left\|\frac{\mathbf{1}_M}{M} - \mathbf{e}_i\right\|$$

$$\le \sum_{j=1}^{k-1}\mathbb{E}\|(\widehat{\mathbf{g}}(\epsilon_j, Z_j) - \mathbf{g}(Z_j))\|_F^2 \cdot \left\|\left(\frac{\mathbf{1}_M}{M} - W^{t(k-j)}\mathbf{e}_i\right)\right\|^2 + \mathbb{E}\|(\widehat{\mathbf{g}}(\epsilon_{k-1}, Z_{k-1}) - \mathbf{g}(Z_{k-1}))\|_F^2 \cdot \left\|\left(\frac{\mathbf{1}_M}{M} - \mathbf{e}_i\right)\right\|^2$$

$$+ 2\mathbb{E}\|\widehat{\mathbf{g}}(\epsilon_{k-1}, Z_{k-1}) - \mathbf{g}(Z_{k-1})\|_F^2 \cdot \left\|\frac{\mathbf{1}_M}{M} - W^t\mathbf{e}_i\right\| \cdot \left\|\frac{\mathbf{1}_M}{M} - \mathbf{e}_i\right\|$$

$$\stackrel{(b)}{\le} M\sigma^2\sum_{j=1}^{k-1}\rho^{2t(k-j)} + M\sigma^2 + 2M\sigma^2\rho^{2t} \le \frac{4M\sigma^2}{1-\rho^{2t}}$$

$$\tag{4}$$

where (a) holds since $\epsilon_i$, $\epsilon_j$ with $i \neq j$ are mutually conditionally independent of each other, (b) holds because of Proposition 1. In addition, note that

$$\mathbb{E}\left\| \sum_{j=1}^{k-1} \mathbf{g}(Z_j)\left( \frac{\mathbf{1}_M}{M} - W^{t(k-j)}\mathbf{e}_i \right) + \left( \mathbf{g}(Z_{k-1})\left( \frac{\mathbf{1}_M}{M} - \mathbf{e}_i \right) \right) \right\|^2$$

$$= \sum_{j=1}^{k-1} \mathbb{E}\left\| \mathbf{g}(Z_j)\left( \frac{\mathbf{1}_M}{M} - W^{t(k-j)}\mathbf{e}_i \right) \right\|^2 + \sum_{j \neq j'} \mathbb{E}\left\langle \mathbf{g}(Z_j)\left( \frac{\mathbf{1}_M}{M} - W^{t(k-j)}\mathbf{e}_i \right), \mathbf{g}(Z_{j'})\left( \frac{\mathbf{1}_M}{M} - W^{t(k-j')}\mathbf{e}_i \right) \right\rangle$$

$$+ 2 \sum_{j \neq k-1} \mathbb{E}\left\langle \mathbf{g}(Z_j)\left( \frac{\mathbf{1}_M}{M} - W^{t(k-j)}\mathbf{e}_i \right), \mathbf{g}(Z_{k-1})\left( \frac{\mathbf{1}_M}{M} - \mathbf{e}_i \right) \right\rangle + \mathbb{E}\left\| \mathbf{g}(Z_{k-1})\left( \frac{\mathbf{1}_M}{M} - \mathbf{e}_i \right) \right\|^2$$

$$:= \mathbf{I}_1 + \mathbf{I}_2 + \mathbf{I}_3 + \mathbf{I}_4$$

$$(5)$$

Now we try to bound these terms separately. Note that

$$\mathbf{I}_1 + \mathbf{I}_4 \leq \sum_{j=1}^{k-1} \mathbb{E}\left\| \mathbf{g}(Z_j) \right\|^2 \cdot \left\| \left( \frac{\mathbf{1}_M}{M} - W^{t(k-j)}\mathbf{e}_i \right) \right\|^2 + \mathbb{E}\|\mathbf{g}(Z_{k-1})\|^2 \left\| \frac{\mathbf{1}_M}{M} - \mathbf{e}_i \right\|^2$$

$$\overset{(a)}{\leq} 3 \sum_{j=1}^{k-1} \left( \sum_{h=1}^{M} \mathbb{E}L^2 \left\| \frac{\sum_{i'=1}^{M} \mathbf{z}_j^{i'}}{M} - \mathbf{z}_j^h \right\|^2 + \mathbb{E}\left\| \mathbf{g}\left( \frac{Z_j \mathbf{1}_M}{M} \right) \mathbf{1}_M^\top \right\|^2 \right) \cdot \left\| \left( \frac{\mathbf{1}_M}{M} - W^{t(k-j)}\mathbf{e}_i \right) \right\|^2$$

$$+ 3 \sum_{h=1}^{M} \left( \mathbb{E}L^2 \left\| \frac{\sum_{i'=1}^{M} \mathbf{z}_{k-1}^{i'}}{M} - \mathbf{z}_{k-1}^h \right\|^2 + \mathbb{E}\left\| \mathbf{g}\left( \frac{Z_{k-1} \mathbf{1}_M}{M} \right) \mathbf{1}_M^\top \right\|^2 \right)$$

where (a) comes from the Proposition 2.

Furthermore, we note that

$$\mathbf{I}_2 \leq \sum_{j \neq j'} \mathbb{E}\left\| \mathbf{g}(Z_j)\left( \frac{\mathbf{1}_M}{M} - W^{t(k-j)}\mathbf{e}_i \right) \right\| \cdot \left\| \mathbf{g}(Z_{j'})\left( \frac{\mathbf{1}_M}{M} - W^{t(k-j')}\mathbf{e}_i \right) \right\|$$

$$\leq \sum_{j \neq j'} \mathbb{E}\|\mathbf{g}(Z_j)\| \left\| \frac{\mathbf{1}_M}{M} - W^{t(k-j)}\mathbf{e}_i \right\| \|\mathbf{g}(Z_{j'})\| \left\| \frac{\mathbf{1}_M}{M} - W^{t(k-j')}\mathbf{e}_i \right\|$$

$$\leq \sum_{j \neq j'} \frac{1}{2}\mathbb{E}\|\mathbf{g}(Z_j)\|^2 \left\| \frac{\mathbf{1}_M}{M} - W^{t(k-j)}\mathbf{e}_i \right\| \left\| \frac{\mathbf{1}_M}{M} - W^{t(k-j')}\mathbf{e}_i \right\|$$

$$+ \sum_{j \neq j'} \frac{1}{2}\mathbb{E}\|\mathbf{g}(Z_{j'})\|^2 \left\| \frac{\mathbf{1}_M}{M} - W^{t(k-j)}\mathbf{e}_i \right\| \left\| \frac{\mathbf{1}_M}{M} - W^{t(k-j')}\mathbf{e}_i \right\|$$

$$\overset{(a)}{\leq} \sum_{j \neq j'} \frac{1}{2}\mathbb{E}\left( \|\mathbf{g}(Z_j)\|^2 + \|\mathbf{g}(Z_{j'})\|^2 \right) \rho^{t(2k-(j+j'))} = \sum_{j \neq j'} \mathbb{E}\|\mathbf{g}(Z_j)\|^2 \rho^{t(2k-(j+j'))}$$

$$\overset{(b)}{\leq} 3 \sum_{j \neq j'} \left( \sum_{h=1}^{M} \mathbb{E}L^2 \left\| \frac{\sum_{i'=1}^{M} \mathbf{z}_j^i}{M} - \mathbf{z}_j^h \right\|^2 + \mathbb{E}\left\| \mathbf{g}\left( \frac{Z_j \mathbf{1}_M}{M} \right) \mathbf{1}_M^\top \right\|^2 \right) \rho^{t(2k-(j+j'))}$$

$$= 6 \sum_{j=1}^{k-1} \left( \sum_{h=1}^{M} \mathbb{E}L^2 \left\| \frac{\sum_{i'=1}^{M} \mathbf{z}_j^i}{M} - \mathbf{z}_j^h \right\|^2 + \mathbb{E}\left\| \mathbf{g}\left( \frac{Z_j \mathbf{1}_M}{M} \right) \mathbf{1}_M^\top \right\|^2 \right) \sum_{j'=j+1}^{k-1} \rho^{t(2k-(j+j'))}$$

$$\leq 6 \sum_{j=1}^{k-1} \left( \sum_{h=1}^{M} \mathbb{E}L^2 \left\| \frac{\sum_{i'=1}^{M} \mathbf{z}_j^i}{M} - \mathbf{z}_j^h \right\|^2 + \mathbb{E}\left\| \mathbf{g}\left( \frac{Z_j \mathbf{1}_M}{M} \right) \mathbf{1}_M^\top \right\|^2 \right) \frac{\rho^{t(k-j)}}{1 - \rho^t}$$

where (a) holds by Proposition 1, (b) comes from Proposition 2.

Following the similar analysis of bounding $\mathbf{I}_2$, we can bound $\mathbf{I}_3$ as in the following:

$$
\begin{aligned}
\mathbf{I}_3 &= 2 \sum_{j=1}^{k-2} \mathbb{E} \left\langle \mathbf{g}(Z_j) \left( \frac{\mathbf{1}_M}{M} - W^{t(k-j)} \mathbf{e}_i \right), \mathbf{g}(Z_{k-1}) \left( \frac{\mathbf{1}_M}{M} - \mathbf{e}_i \right) \right\rangle \\
&\leq \sum_{j=1}^{k-2} \mathbb{E} \left\| \mathbf{g}(Z_j) \right\|^2 \cdot \left\| \frac{\mathbf{1}_M}{M} - W^{t(k-j)} \mathbf{e}_i \right\|^2 + \mathbb{E} \left\| \mathbf{g}(Z_{k-1}) \right\|^2 \cdot \left\| \frac{\mathbf{1}_M}{M} - \mathbf{e}_i \right\|^2 \\
&\leq 3 \sum_{j=1}^{k-2} \left( \sum_{h=1}^{M} \mathbb{E} L^2 \left\| \frac{\sum_{i'=1}^{M} \mathbf{z}_j^i}{M} - \mathbf{z}_j^h \right\|^2 + \mathbb{E} \left\| \mathbf{g} \left( \frac{Z_j \mathbf{1}_M}{M} \right) \mathbf{1}_M^\top \right\|^2 \right) \rho^{2t(k-j)} + \mathbb{E} \left\| \mathbf{g} \left( \frac{Z_{k-1} \mathbf{1}_M}{M} \right) \mathbf{1}_M^\top \right\|^2
\end{aligned}
$$

Using the bound of $\mathbf{I}_1$, $\mathbf{I}_2$, $\mathbf{I}_3$, $\mathbf{I}_4$ and by (4) and (5), we know that,

$$
\begin{aligned}
\text{RHS of (3)} &\leq \frac{2L^2 \eta^2}{M} \left[ \frac{4M\sigma^2}{1-\rho^{2t}} + 6 \sum_{j=1}^{k-1} \left( \sum_{h=1}^{M} \mathbb{E} L^2 \left\| \frac{\sum_{i'=1}^{M} \mathbf{z}_j^i}{M} - \mathbf{z}_j^h \right\|^2 \right) \left( \frac{\rho^{t(k-j)}}{1-\rho^t} + \rho^{2t(k-j)} \right) \right. \\
&\quad + 6 \sum_{j=1}^{k-1} \mathbb{E} \left\| \mathbf{g} \left( \frac{Z_j \mathbf{1}_M}{M} \right) \mathbf{1}_M^\top \right\|^2 \left( \rho^{2t(k-j)} + \frac{\rho^{t(k-j)}}{1-\rho^t} \right) \\
&\quad \left. + 4 \sum_{h=1}^{M} \mathbb{E} L^2 \left\| \frac{\sum_{i'=1}^{M} \mathbf{z}_{k-1}^{i'}}{M} - \mathbf{z}_{k-1}^h \right\|^2 + 4\mathbb{E} \left\| \mathbf{g} \left( \frac{Z_{k-1} \mathbf{1}_M}{M} \right) \mathbf{1}_M^\top \right\|^2 \right]
\end{aligned}
$$

(6)

where (a) comes from Proposition 1.

Define $\lambda_k = \frac{1}{M} \sum_{i=1}^{M} \left\| \mathbf{z}_k^i - \bar{\mathbf{z}}_k \right\|^2$. By (6) and (3), then we have

$$
\begin{aligned}
\mathbb{E}[\lambda_k] &\leq \frac{8\eta^2 M \sigma^2}{1-\rho^{2t}} + 12\eta^2 \sum_{j=1}^{k-1} \mathbb{E} \left\| \mathbf{g} \left( \frac{Z_j \mathbf{1}_M}{M} \right) \mathbf{1}_M^\top \right\|^2 \left( \rho^{2t(k-j)} + \frac{\rho^{t(k-j)}}{1-\rho^t} \right) \\
&\quad + 12\eta^2 M L^2 \sum_{j=1}^{k-1} \mathbb{E}[\lambda_j] \left( \rho^{2t(k-j)} + \frac{\rho^{t(k-j)}}{1-\rho^t} \right) + 8\eta^2 M L^2 \mathbb{E}[\lambda_{k-1}] + 8\eta^2 \mathbb{E} \left\| \mathbf{g} \left( \frac{Z_{k-1} \mathbf{1}_M}{M} \right) \mathbf{1}_M^\top \right\|^2.
\end{aligned}
$$

(7)

Define $\lambda_{-1} = 0$. Summing over $k = 0, \ldots, N-1$ on both sides of (7) yield

$$
\begin{aligned}
\sum_{k=0}^{N-1} \mathbb{E}[\lambda_k] &\leq 12\eta^2 \sum_{k=0}^{N-1} \sum_{j=1}^{k-1} \mathbb{E} \left\| \mathbf{g} \left( \frac{Z_j \mathbf{1}_M}{M} \right) \mathbf{1}_M^\top \right\|^2 \left( \rho^{2t(k-j)} + \frac{\rho^{t(k-j)}}{1-\rho^t} \right) + 8\eta^2 \sum_{k=0}^{N-1} \mathbb{E} \left\| \mathbf{g} \left( \frac{Z_{k-1} \mathbf{1}_M}{M} \right) \mathbf{1}_M^\top \right\|^2 \\
&\quad + 12\eta^2 M L^2 \sum_{k=0}^{N-1} \sum_{j=1}^{k-1} \mathbb{E}[\lambda_j] \left( \rho^{2t(k-j)} + \frac{\rho^{t(k-j)}}{1-\rho^t} \right) + 8\eta^2 M L^2 \sum_{k=0}^{N-1} \mathbb{E}[\lambda_{k-1}] + \frac{8\eta^2 M \sigma^2 N}{1-\rho^{2t}} \\
&\leq \frac{8\eta^2 M \sigma^2 N}{1-\rho^{2t}} + 12\eta^2 \sum_{k=0}^{N-1} \mathbb{E} \left\| \mathbf{g} \left( \frac{Z_k \mathbf{1}_M}{M} \right) \mathbf{1}_M^\top \right\|^2 \left( \sum_{i=0}^{\infty} \rho^{2ti} + \frac{\sum_{i=0}^{\infty} \rho^{ti}}{1-\rho^t} \right) + 8\eta^2 \sum_{k=0}^{N-1} \mathbb{E} \left\| \mathbf{g} \left( \frac{Z_{k-1} \mathbf{1}_M}{M} \right) \mathbf{1}_M^\top \right\|^2 \\
&\quad + 12\eta^2 M L^2 \sum_{k=0}^{N-1} \mathbb{E}[\lambda_k] \left( \sum_{i=0}^{\infty} \rho^{2ti} + \frac{\sum_{i=0}^{\infty} \rho^{ti}}{1-\rho^t} \right) + 8\eta^2 M L^2 \sum_{k=0}^{N-1} \mathbb{E}[\lambda_{k-1}] + \frac{8\eta^2 M \sigma^2 N}{1-\rho^{2t}} \\
&\leq \frac{8\eta^2 M \sigma^2 N}{1-\rho^{2t}} + \frac{32\eta^2}{(1-\rho^t)^2} \sum_{k=0}^{N-1} \mathbb{E} \left\| \mathbf{g} \left( \frac{Z_k \mathbf{1}_M}{M} \right) \mathbf{1}_M^\top \right\|^2 + \frac{32\eta^2 M L^2}{(1-\rho^t)^2} \sum_{k=0}^{N-1} \mathbb{E}[\lambda_k]
\end{aligned}
$$

(8)

Rearrange the terms in (8), and we have

$$\frac{1}{N}\sum_{k=0}^{N-1}\mathbb{E}\left[\lambda_k\right] \leq \frac{8\eta^2 M\sigma^2}{(1-\rho^{2t})\left(1-\frac{32\eta^2 ML^2}{(1-\rho^t)^2}\right)} + \frac{32\eta^2}{(1-\rho^t)^2\left(1-\frac{32\eta^2 ML^2}{(1-\rho^t)^2}\right)} \cdot \frac{1}{N}\sum_{k=0}^{N-1}\mathbb{E}\left\|\mathbf{g}\left(\frac{Z_k \mathbf{1}_M}{M}\right)\mathbf{1}_M^\top\right\|^2$$

$$= \frac{8\eta^2 M\sigma^2}{(1-\rho^{2t})\left(1-\frac{32\eta^2 ML^2}{(1-\rho^t)^2}\right)} + \frac{32\eta^2 M}{(1-\rho^t)^2\left(1-\frac{32\eta^2 ML^2}{(1-\rho^t)^2}\right)} \cdot \frac{1}{N}\sum_{k=0}^{N-1}\mathbb{E}\left\|\mathbf{g}\left(\frac{Z_k \mathbf{1}_M}{M}\right)\right\|^2$$

(9)

Combining (9) and (3) suffices to prove the lemma.

$\square$

## A.3  Proof of Lemmas

### A.3.1  Proof of Lemma 1

**Lemma 1 restated**  By taking $\eta = \min\left(\frac{1-\rho^t}{\sqrt{32cML}}, \frac{\sqrt{1-\rho^{2t}}}{4m^{1/4}M^{3/4}L}\right)$ with $c \geq 2$, we have

$$\frac{1}{N}\sum_{k=0}^{N-1}\mathbb{E}\left\|\frac{1}{M}\mathbf{g}(Z_k)\mathbf{1}_M - \mathbf{g}(\bar{\mathbf{z}}_k)\right\|^2 \leq \frac{\sigma^2}{\sqrt{mM}} + \frac{1}{c-1}\cdot\frac{1}{N}\sum_{k=0}^{N-1}\mathbb{E}\left\|\mathbf{g}\left(\bar{\mathbf{z}}_k\right)\right\|^2.$$

*Proof.* By $\eta^2 \leq \frac{(1-\rho^t)^2}{32cML^2} \leq \frac{(1-\rho^t)^2}{64ML^2}$, we know that $1 - \frac{32\eta^2 ML^2}{(1-\rho^t)^2} \geq \frac{1}{2}$. In addition, since $\eta^2 \leq \frac{1-\rho^{2t}}{16\sqrt{mM^3}L^2}$, we know that

$$\frac{8\eta^2 ML^2\sigma^2}{(1-\rho^{2t})\left(1-\frac{32\eta^2 ML^2}{(1-\rho^t)^2}\right)} \leq \frac{16\eta^2 ML^2\sigma^2}{1-\rho^{2t}} \leq \frac{\sigma^2}{\sqrt{mM}} \tag{10}$$

Note that $\frac{32\eta^2 ML^2}{(1-\rho^t)^2\left(1-\frac{32\eta^2 ML^2}{(1-\rho^t)^2}\right)} = \frac{32ML^2}{\frac{(1-\rho^t)^2}{\eta^2}-32ML^2}$ is monotonically increasing in terms of $\eta^2$, and $\eta^2 \leq \frac{(1-\rho^t)^2}{32cML^2}$, and hence we have

$$\frac{32\eta^2 ML^2}{(1-\rho^t)^2\left(1-\frac{32\eta^2 ML^2}{(1-\rho^t)^2}\right)} \leq \frac{1}{c-1}. \tag{11}$$

Combining (10), (11) and Lemma 4 suffices to show the result.  $\square$

### A.3.2  Proof of Lemma 2

**Lemma 2 restated**  The following inequality holds for the stochastic gradient:

$$\frac{1}{M}\sum_{i=1}^{M}\mathbb{E}\left[\left\|\frac{1}{M}\widehat{\mathbf{g}}(\epsilon_{k-1},Z_{k-1})\mathbf{1}_M - \widehat{\mathbf{g}}(\epsilon_{k-1},Z_{k-1})\mathbf{e}_i\right\|\right] \leq \frac{2\sigma}{\sqrt{mM}} + 2\mathbb{E}\left[\frac{1}{M}\sum_{i=1}^{M}\left\|\mathbf{g}(\mathbf{z}_{k-1}^i)-\mathbf{g}(\bar{\mathbf{z}}_{k-1})\right\|\right].$$

*Proof.*

$$\frac{1}{M}\sum_{i=1}^{M}\mathbb{E}\left[\left\|\frac{1}{M}\widehat{\mathbf{g}}(\epsilon_{k-1},Z_{k-1})\mathbf{1}_M-\widehat{\mathbf{g}}(\epsilon_{k-1},Z_{k-1})\mathbf{e}_i\right\|\right]=\frac{1}{M}\sum_{i=1}^{M}\mathbb{E}\left[\left\|\sum_{j=1}^{M}\frac{\widehat{\mathbf{g}}(\epsilon_{k-1}^j,\mathbf{z}_{k-1}^i)-\widehat{\mathbf{g}}(\epsilon_{k-1}^i,\mathbf{z}_{k-1}^j)}{M}\right\|\right]$$

$$=\frac{1}{M}\sum_{i=1}^{M}\mathbb{E}\left[\left\|\sum_{j=1}^{M}\frac{\widehat{\mathbf{g}}(\epsilon_{k-1}^j,\mathbf{z}_{k-1}^j)-\mathbf{g}(\mathbf{z}_{k-1}^j)+\mathbf{g}(\mathbf{z}_{k-1}^j)-\mathbf{g}(\bar{\mathbf{z}}_{k-1})+\mathbf{g}(\bar{\mathbf{z}}_{k-1})-\mathbf{g}(\mathbf{z}_{k-1}^i)+\mathbf{g}(\mathbf{z}_{k-1}^i)-\widehat{\mathbf{g}}(\epsilon_{k-1}^i,\mathbf{z}_{k-1}^i)}{M}\right\|\right]$$

$$\stackrel{(a)}{=}\frac{1}{M}\sum_{i=1}^{M}\mathbb{E}\left[\left\|\sum_{j=1}^{M}\frac{\epsilon_{k-1}^j+\mathbf{g}(\mathbf{z}_{k-1}^j)-\mathbf{g}(\bar{\mathbf{z}}_{k-1})+\mathbf{g}(\bar{\mathbf{z}}_{k-1})-\mathbf{g}(\mathbf{z}_{k-1}^i)-\epsilon_{k-1}^i}{M}\right\|\right]$$

$$\stackrel{(b)}{\leq}\frac{1}{M}\sum_{i=1}^{M}\mathbb{E}\left[\left\|\frac{1}{M}\sum_{j=1}^{M}\left(\epsilon_{k-1}^j-\epsilon_{k-1}^i\right)\right\|+\frac{1}{M}\sum_{j=1}^{M}\left\|\mathbf{g}(\mathbf{z}_{k-1}^j)-\mathbf{g}(\bar{\mathbf{z}}_{k-1})\right\|+\left\|\mathbf{g}(\mathbf{z}_{k-1}^i)-\mathbf{g}(\bar{\mathbf{z}}_{k-1})\right\|\right]$$

$$\leq\frac{2\sigma}{\sqrt{mM}}+2\mathbb{E}\left[\frac{1}{M}\sum_{i=1}^{M}\left\|\mathbf{g}(\mathbf{z}_{k-1}^i)-\mathbf{g}(\bar{\mathbf{z}}_{k-1})\right\|\right]$$

where (a) holds by the definition of $\epsilon_{k-1}^i$ and $\epsilon_{k-1}^j$, (b) holds by the triangle inequality of norm, (c) holds since $\epsilon_{k-1}^j$ and $\epsilon_{k-1}^i$ with $i\neq j$ are conditionally mutually independent of each other and the fact that $\mathbb{E}\|\mathbf{x}\|\leq\sqrt{\mathbb{E}\|\mathbf{x}\|^2}$. $\qquad\qquad\square$

### A.3.3 Proof of Lemma 3

**Lemma 3 restated** Define $\mu_k=\frac{1}{M}\sum_{i=1}^{M}\left\|\mathbf{g}(\mathbf{z}_k^i)-\mathbf{g}(\bar{\mathbf{z}}_k)\right\|$. Suppose $\eta<\frac{1}{4L}$ and $t\geq\log_{\frac{1}{\rho}}\left(1+\frac{M\sqrt{mM}G^2+\sigma^2}{4\sigma}\right)$. We have

$$\frac{1}{N}\sum_{k=0}^{N-1}\mathbb{E}\left[\mu_k\right]<\frac{8\eta L\sigma}{\sqrt{mM}(1-4\eta L)}.$$

*Proof.* Define $\mathbf{e}_i=[0,\ldots,1,\ldots,0]^\top$, where 1 appears at the $i$-th coordinate and the rest entries are all zeros. Then we have

$$\mu_k\stackrel{(a)}{\leq}\frac{L}{M}\sum_{i=1}^{M}\left\|\mathbf{z}_k^i-\bar{\mathbf{z}}_k\right\|$$

$$\stackrel{(b)}{=}\frac{L}{M}\sum_{i=1}^{M}\left\|\frac{1}{M}X_{k-1}\mathbf{1}_M-\frac{1}{M}\eta\widehat{\mathbf{g}}(\epsilon_{k-1},Z_{k-1})\mathbf{1}_M-\left(X_{k-1}W^t\mathbf{e}_i-\eta\widehat{\mathbf{g}}(\epsilon_{k-1},Z_{k-1})\mathbf{e}_i\right)\right\|$$

$$=\frac{L}{M}\sum_{i=1}^{M}\left\|\frac{1}{M}\left(X_0-\eta\sum_{j=1}^{k-1}\widehat{\mathbf{g}}(\epsilon_j,Z_j)-\eta\widehat{\mathbf{g}}(\epsilon_{k-1},Z_{k-1})\right)\mathbf{1}_M\right.$$

$$\left.-\left(X_0W^{tk}-\eta\sum_{j=1}^{k-1}\widehat{\mathbf{g}}(\epsilon_j,Z_j)W^{(k-j)t}-\eta\widehat{\mathbf{g}}(\epsilon_{k-1},Z_{k-1})\right)\mathbf{e}_i\right\|$$

$$=\frac{L}{M}\sum_{i=1}^{M}\left\|\eta\sum_{j=1}^{k-1}\widehat{\mathbf{g}}(\epsilon_j,Z_j)\left(\frac{1}{M}\mathbf{1}_M-W^{(k-j)t}\mathbf{e}_i\right)\right\|+\frac{2\eta L}{M}\sum_{i=1}^{M}\left\|\frac{1}{M}\widehat{\mathbf{g}}(\epsilon_{k-1},Z_{k-1})\mathbf{1}_M-\widehat{\mathbf{g}}(\epsilon_{k-1},Z_{k-1})\mathbf{e}_i\right\|$$

$$\stackrel{(c)}{\leq}\frac{L\eta}{M}\sum_{i=1}^{M}\left\|\sum_{j=1}^{k-1}\widehat{\mathbf{g}}(\epsilon_j,Z_j)\rho^{(k-j)t}\right\|+\frac{2\eta L}{M}\sum_{i=1}^{M}\left\|\frac{1}{M}\widehat{\mathbf{g}}(\epsilon_{k-1},Z_{k-1})\mathbf{1}_M-\widehat{\mathbf{g}}(\epsilon_{k-1},Z_{k-1})\mathbf{e}_i\right\|$$

$$\leq\frac{L\eta}{M}\sum_{i=1}^{M}\sum_{j=1}^{k-1}\rho^{(k-j)t}\left\|\widehat{\mathbf{g}}(\epsilon_j,Z_j)\right\|_F+\frac{2\eta L}{M}\sum_{i=1}^{M}\left\|\frac{1}{M}\widehat{\mathbf{g}}(\epsilon_{k-1},Z_{k-1})\mathbf{1}_M-\widehat{\mathbf{g}}(\epsilon_{k-1},Z_{k-1})\mathbf{e}_i\right\|$$

$$\tag{12}$$

where (a) holds by the $L$-Lipschitz continuity of $\mathbf{g}$, (b) holds by the update and $W\mathbf{1}_M = \mathbf{1}_M$, (c) holds by Proposition 1.

Taking expectation over $\epsilon_0, \ldots, \epsilon_{k-1}$ on both sides of (12) yields

$$
\mathbb{E}\left[\mu_k\right] \overset{(a)}{\leq} \frac{\eta L \rho^t}{1-\rho^t} M \sqrt{G^2 + \frac{\sigma^2}{m}} + \frac{2\eta L}{M} \sum_{i=1}^{M} \mathbb{E}\left[\left\|\frac{1}{M}\widehat{\mathbf{g}}(\epsilon_{k-1}, Z_{k-1})\mathbf{1}_M - \widehat{\mathbf{g}}(\epsilon_{k-1}, Z_{k-1})\mathbf{e}_i\right\|\right]
$$

$$
\overset{(b)}{\leq} \frac{\eta L \rho^t}{1-\rho^t} M \sqrt{G^2 + \frac{\sigma^2}{m}} + 2\eta L \left(2\mathbb{E}\left[\mu_{k-1}\right] + \frac{2\sigma}{\sqrt{Mm}}\right)
$$

$$
= \frac{\eta L \rho^t}{1-\rho^t} M \sqrt{G^2 + \frac{\sigma^2}{m}} + 4\eta L \left(\mathbb{E}\left[\mu_{k-1}\right] + \frac{\sigma}{\sqrt{mM}}\right)
$$

(13)

where (a) holds since $\mathbb{E}\left[\mathbf{g}(\mathbf{x};\xi)\right] = \mathbf{g}(\mathbf{x})$, $\mathbb{E}\left\|\mathbf{g}(\mathbf{x};\xi) - \mathbf{g}(\mathbf{x})\right\|^2 \leq \sigma^2$, $\|\mathbf{g}(\mathbf{x})\| \leq G$, and (b) holds by invoking Lemma 2.

Define $\delta = \frac{\rho^t}{4(1-\rho^t)} M \sqrt{G^2 + \frac{\sigma^2}{m}}$. By taking $t \geq \log_{\frac{1}{\rho}}\left(1 + \frac{M\sqrt{mM G^2 + \sigma^2}}{4\sigma}\right)$, we can show that $\delta \leq \frac{\sigma}{\sqrt{mM}}$. Hence we can rewrite (13) as

$$
\mathbb{E}\left[\mu_k\right] \leq 4\eta L \left(\mathbb{E}\left[\mu_{k-1}\right] + \frac{2\sigma}{\sqrt{mM}}\right).
$$

Define $b_k = \mathbb{E}\left[\mu_k\right] + \frac{2\sigma}{\sqrt{mM}}$. Then we have $b_k \leq 4\eta L b_{k-1} + \frac{2\sigma}{\sqrt{mM}}$, which implies that $b_k + \frac{\frac{2\sigma}{\sqrt{mM}}}{4\eta L - 1} \leq 4\eta L \left(b_{k-1} + \frac{\frac{2\sigma}{\sqrt{mM}}}{4\eta L - 1}\right)$. Note that $b_0 = \mathbb{E}\left[\mu_0\right] + \frac{2\sigma}{\sqrt{mM}} = \frac{2\sigma}{\sqrt{mM}}$ and $4\eta L < 1$, and hence we have

$$
\frac{1}{N} \sum_{k=0}^{N-1}\left(b_k + \frac{\frac{2\sigma}{\sqrt{mM}}}{4\eta L - 1}\right) \leq \frac{\frac{2\sigma}{\sqrt{mM}}\left(1 + \frac{1}{4\eta L - 1}\right)}{N(1 - 4\eta L)} < 0.
$$

So we know that

$$
\frac{1}{N} \sum_{k=0}^{N-1} \mathbb{E}\left[\mu_k\right] = \frac{1}{N} \sum_{k=0}^{N-1} b_k - \frac{2\sigma}{\sqrt{mM}} < \frac{1}{N} \sum_{k=0}^{N-1} b_k < \frac{\frac{2\sigma}{\sqrt{mM}} \cdot 4\eta L}{1 - 4\eta L} = \frac{8\eta L \sigma}{\sqrt{mM}(1 - 4\eta L)}.
$$

Here completes the proof. $\qquad\square$

## A.4 Main Proof of Theorem 1

*Proof.* Define $\mathbf{1}_M = [1, \ldots, 1]^\top \in \mathbb{R}^{M \times 1}$, $\bar{\mathbf{z}}_k = \frac{1}{M} Z_k \mathbf{1}_M$, $\bar{\mathbf{x}}_k = \frac{1}{M} X_k \mathbf{1}_M$, $\bar{\epsilon}_k = \frac{1}{M} \sum_{i=1}^{M} \epsilon_k^i$. By the property of $W$, we know that $W\mathbf{1}_M = \mathbf{1}_M$.

Noting that for $\forall \mathbf{x} \in \mathcal{X} = \mathbb{R}^d$, we have

$$
\left\|\frac{1}{M} X_k \mathbf{1}_M - \mathbf{x}\right\|^2 = \left\|\frac{1}{M}\left(X_{k-1} W^t - \eta\widehat{\mathbf{g}}(\epsilon_k, Z_k)\right) \cdot \mathbf{1}_M - \mathbf{x}\right\|^2
$$

$$
= \left\|\frac{1}{M}\left(X_{k-1} W^t - \eta\widehat{\mathbf{g}}(\epsilon_k, Z_k)\right) \cdot \mathbf{1}_M - \mathbf{x}\right\|^2 - \left\|\frac{1}{M}\left(X_{k-1} W^t - \eta\widehat{\mathbf{g}}(\epsilon_k, Z_k) - X_k\right) \cdot \mathbf{1}_M\right\|^2
$$

$$
= \|\bar{\mathbf{x}}_{k-1} - \mathbf{x}\|^2 - \|\bar{\mathbf{x}}_{k-1} - \bar{\mathbf{x}}_k\|^2 + 2\left\langle \mathbf{x} - \bar{\mathbf{x}}_k, \frac{1}{M}\eta\widehat{\mathbf{g}}(\epsilon_k, Z_k)\mathbf{1}_M \right\rangle
$$

$$
= \|\bar{\mathbf{x}}_{k-1} - \mathbf{x}\|^2 - \|\bar{\mathbf{x}}_{k-1} - \bar{\mathbf{x}}_k\|^2 + 2\left\langle \mathbf{x} - \bar{\mathbf{z}}_k, \frac{1}{M}\eta\widehat{\mathbf{g}}(\epsilon_k, Z_k)\mathbf{1}_M \right\rangle + 2\left\langle \bar{\mathbf{z}}_k - \bar{\mathbf{x}}_k, \frac{1}{M}\eta\widehat{\mathbf{g}}(\epsilon_k, Z_k)\mathbf{1}_M \right\rangle
$$

$$
= \|\bar{\mathbf{x}}_{k-1} - \mathbf{x}\|^2 - \|\bar{\mathbf{x}}_{k-1} - \bar{\mathbf{z}}_k + \bar{\mathbf{z}}_k - \bar{\mathbf{x}}_k\|^2 + 2\left\langle \mathbf{x} - \bar{\mathbf{z}}_k, \frac{1}{M}\eta\widehat{\mathbf{g}}(\epsilon_k, Z_k)\mathbf{1}_M \right\rangle + 2\left\langle \bar{\mathbf{z}}_k - \bar{\mathbf{x}}_k, \frac{1}{M}\eta\widehat{\mathbf{g}}(\epsilon_k, Z_k)\mathbf{1}_M \right\rangle
$$

$$
= \|\bar{\mathbf{x}}_{k-1} - \mathbf{x}\|^2 - \|\bar{\mathbf{x}}_{k-1} - \bar{\mathbf{z}}_k\|^2 - \|\bar{\mathbf{z}}_k - \bar{\mathbf{x}}_k\|^2 + 2\left\langle \mathbf{x} - \bar{\mathbf{z}}_k, \frac{1}{M}\eta\widehat{\mathbf{g}}(\epsilon_k, Z_k)\mathbf{1}_M \right\rangle
$$

$$
+ 2\left\langle \bar{\mathbf{x}}_k - \bar{\mathbf{z}}_k, \bar{\mathbf{x}}_{k-1} - \frac{1}{M}\eta\widehat{\mathbf{g}}(\epsilon_k, Z_k)\mathbf{1}_M - \bar{\mathbf{z}}_k \right\rangle
$$

(14)

Note that

$$
2\left\langle \mathbf{x}_* - \bar{\mathbf{z}}_k, \frac{1}{M}\eta\widehat{\mathbf{g}}(\epsilon_k, Z_k)\mathbf{1}_M \right\rangle = 2\left\langle \mathbf{x}_* - \bar{\mathbf{z}}_k, \frac{1}{M}\eta\mathbf{g}(Z_k)\mathbf{1}_M \right\rangle + 2\left\langle \mathbf{x}_* - \bar{\mathbf{z}}_k, \frac{1}{M}\eta\sum_{i=1}^{M}\epsilon_k^i \right\rangle
$$

$$
= 2\left\langle \mathbf{x}_* - \bar{\mathbf{z}}_k, \eta\mathbf{g}(\bar{\mathbf{z}}_k) \right\rangle + 2\left\langle \mathbf{x}_* - \bar{\mathbf{z}}_k, \eta\frac{1}{M}\sum_{i=1}^{M}\left(\mathbf{g}(\mathbf{z}_k^i) - \mathbf{g}(\bar{\mathbf{z}}_k)\right) \right\rangle + 2\left\langle \mathbf{x}_* - \bar{\mathbf{z}}_k, \frac{1}{M}\eta\sum_{i=1}^{M}\epsilon_k^i \right\rangle
$$

$$
\overset{(a)}{\le} 2\left\langle \mathbf{x}_* - \bar{\mathbf{z}}_k, \eta\frac{1}{M}\sum_{i=1}^{M}\left(\mathbf{g}(\mathbf{z}_k^i) - \mathbf{g}(\bar{\mathbf{z}}_k)\right) \right\rangle + 2\left\langle \mathbf{x}_* - \bar{\mathbf{z}}_k, \frac{1}{M}\eta\sum_{i=1}^{M}\epsilon_k^i \right\rangle
$$

$$
\overset{(b)}{\le} 2\eta D \left\| \frac{1}{M}\sum_{i=1}^{M}\left(\mathbf{g}(\mathbf{z}_k^i) - \mathbf{g}(\bar{\mathbf{z}}_k)\right) \right\| + 2\left\langle \mathbf{x}_* - \bar{\mathbf{z}}_k, \frac{1}{M}\eta\sum_{i=1}^{M}\epsilon_k^i \right\rangle
$$

$$
\le 2\eta D\frac{1}{M}\sum_{i=1}^{M}\left\|\mathbf{g}(\mathbf{z}_k^i) - \mathbf{g}(\bar{\mathbf{z}}_k)\right\| + 2\left\langle \mathbf{x}_* - \bar{\mathbf{z}}_k, \frac{1}{M}\eta\sum_{i=1}^{M}\epsilon_k^i \right\rangle
$$

(15)

where (a) holds since $\langle \mathbf{x}_* - \bar{\mathbf{z}}_k, \eta\mathbf{g}(\bar{\mathbf{z}}_k) \rangle \le 0$, (b) holds by Cauchy-Schwarz inequality and $\|\bar{\mathbf{z}}_k - \mathbf{x}_*\| \le D$. Note that

$$
2\left\langle \bar{\mathbf{x}}_k - \bar{\mathbf{z}}_k, \bar{\mathbf{x}}_{k-1} - \frac{1}{M}\eta\widehat{\mathbf{g}}(\epsilon_k, Z_k)\mathbf{1}_M - \bar{\mathbf{z}}_k \right\rangle
$$

$$
= 2\left\langle \bar{\mathbf{x}}_k - \bar{\mathbf{z}}_k, \bar{\mathbf{x}}_{k-1} - \frac{1}{M}\eta\widehat{\mathbf{g}}(\epsilon_{k-1}, Z_{k-1})\mathbf{1}_M - \bar{\mathbf{z}}_k \right\rangle
$$

$$
+ 2\left\langle \bar{\mathbf{x}}_k - \bar{\mathbf{z}}_k, \frac{1}{M}\eta\widehat{\mathbf{g}}(\epsilon_k, Z_k)\mathbf{1}_M - \frac{1}{M}\eta\widehat{\mathbf{g}}(\epsilon_{k-1}, Z_{k-1})\mathbf{1}_M \right\rangle
$$

$$
\overset{(a)}{=} 2\left\langle \bar{\mathbf{x}}_k - \bar{\mathbf{z}}_k, \frac{1}{M}\eta\widehat{\mathbf{g}}(\epsilon_k, Z_k)\mathbf{1}_M - \frac{1}{M}\eta\widehat{\mathbf{g}}(\epsilon_{k-1}, Z_{k-1})\mathbf{1}_M \right\rangle
$$

$$
\overset{(b)}{\le} 2\left\| \left( \bar{\mathbf{x}}_{k-1} - \frac{1}{M}\eta\widehat{\mathbf{g}}(\epsilon_k, Z_k)\mathbf{1}_M \right) - \left( \bar{\mathbf{x}}_{k-1} - \frac{1}{M}\eta\widehat{\mathbf{g}}(\epsilon_{k-1}, Z_{k-1})\mathbf{1}_M \right) \right\|
$$

$$
\cdot \left\| \frac{1}{M}\eta\widehat{\mathbf{g}}(\epsilon_k, Z_k)\mathbf{1}_M - \frac{1}{M}\eta\widehat{\mathbf{g}}(\epsilon_{k-1}, Z_{k-1})\mathbf{1}_M \right\| \le 2\eta^2 \left\| \frac{1}{M}\widehat{\mathbf{g}}(\epsilon_k, Z_k)\mathbf{1}_M - \frac{1}{M}\widehat{\mathbf{g}}(\epsilon_{k-1}, Z_{k-1})\mathbf{1}_M \right\|^2
$$

$$
\overset{(c)}{\le} 6\eta^2 \left\| \frac{1}{M}\left(\mathbf{g}(Z_k) - \mathbf{g}(Z_{k-1})\right)\mathbf{1}_M \right\|^2 + 6\eta^2\|\bar{\epsilon}_k\|^2 + 6\eta^2\|\bar{\epsilon}_{k-1}\|^2
$$

$$
\overset{(d)}{\le} 18\eta^2 \left\| \frac{1}{M}\mathbf{g}(Z_k)\mathbf{1}_M - \mathbf{g}(\bar{\mathbf{z}}_k) \right\|^2 + 18\eta^2\left\|\mathbf{g}(\bar{\mathbf{z}}_k) - \mathbf{g}(\bar{\mathbf{z}}_{k-1})\right\|^2 + 18\eta^2\left\| \frac{1}{M}\mathbf{g}(Z_{k-1})\mathbf{1}_M - \mathbf{g}(\bar{\mathbf{z}}_{k-1}) \right\|^2
$$

$$
+ 6\eta^2\|\bar{\epsilon}_k\|^2 + 6\eta^2\|\bar{\epsilon}_{k-1}\|^2
$$

$$
\overset{(e)}{\le} 18\eta^2 \left\| \frac{1}{M}\mathbf{g}(Z_{k-1})\mathbf{1}_M - \mathbf{g}(\bar{\mathbf{z}}_{k-1}) \right\|^2 + 18\eta^2 \left\| \frac{1}{M}\mathbf{g}(Z_k)\mathbf{1}_M - \mathbf{g}(\bar{\mathbf{z}}_k) \right\|^2 + 18\eta^2 L^2 \left\|\bar{\mathbf{z}}_{k-1} - \bar{\mathbf{z}}_k\right\|^2
$$

$$
+ 6\eta^2\|\bar{\epsilon}_{k-1}\|^2 + 6\eta^2\|\bar{\epsilon}_k\|^2
$$

(16)

where (a) holds by the update of the algorithm and the fact that $W^t\mathbf{1}_M = \mathbf{1}_M$, (b) holds by utilizing Cauchy-Schwarz inequality and, (c) and (d) hold since $\|\mathbf{a} + \mathbf{b} + \mathbf{c}\|^2 \le 3\|\mathbf{a}\|^2 + 3\|\mathbf{b}\|^2 + 3\|\mathbf{c}\|^2$, (e) holds by the $L$-Lipschitz continuity of $\mathbf{g}$.

Note that

$$
\begin{aligned}
\eta^2 \left\| \mathbf{g}\left(\frac{1}{M} Z_k \mathbf{1}_M\right) \right\|^2 &= \left\| \bar{\mathbf{z}}_k - (\bar{\mathbf{z}}_k - \eta \mathbf{g}(\bar{\mathbf{z}}_k)) \right\|^2 = \left\| \bar{\mathbf{z}}_k - \bar{\mathbf{x}}_k + (\bar{\mathbf{x}}_k - (\bar{\mathbf{z}}_k - \eta \mathbf{g}(\bar{\mathbf{z}}_k))) \right\|^2 \\
&\leq 2\left\| \bar{\mathbf{z}}_k - \bar{\mathbf{x}}_k \right\|^2 + 2\left\| (\bar{\mathbf{x}}_k - (\bar{\mathbf{z}}_k - \eta \mathbf{g}(\bar{\mathbf{z}}_k))) \right\|^2 \\
&= 2\left\| \bar{\mathbf{z}}_k - \bar{\mathbf{x}}_k \right\|^2 + 2\left\| \left( \frac{1}{M}\left( X_{k-1}W - \eta \widehat{\mathbf{g}}(\epsilon_k, Z_k) \right)\mathbf{1}_M - (\bar{\mathbf{z}}_k - \eta\mathbf{g}(\bar{\mathbf{z}}_k)) \right) \right\|^2 \\
&\leq 2\left\| \bar{\mathbf{z}}_k - \bar{\mathbf{x}}_k \right\|^2 + 4\left\| \frac{1}{M}\left(X_{k-1}W - Z_k\right)\mathbf{1}_M \right\|^2 + 4\eta^2 \left\| \frac{1}{M}\sum_{i=1}^{M}\left( \widehat{\mathbf{g}}(\epsilon_k, Z_k) - \mathbf{g}(\bar{\mathbf{z}}_k) \right) \right\|^2 \\
&\leq 2\left\| \bar{\mathbf{z}}_k - \bar{\mathbf{x}}_k \right\|^2 + 4\left\| \bar{\mathbf{x}}_{k-1} - \mathbf{z}_k \right\|^2 + 4\eta^2 \left\| \frac{1}{M}\sum_{i=1}^{M}\left( \mathbf{g}(\mathbf{z}_k^i) - \mathbf{g}(\bar{\mathbf{z}}_k) \right) \right\|^2 + \frac{4\eta^2}{M^2}\sum_{i=1}^{M}\|\epsilon_k^i\|^2
\end{aligned}
\tag{17}
$$

Taking $\mathbf{x} = \mathbf{x}_*$ in (14), combining (15), (16) and noting that $W\mathbf{1}_M = \mathbf{1}_M$ yield

$$
\begin{aligned}
\left\| \bar{\mathbf{x}}_k - \mathbf{x}_* \right\|^2 &\leq \left\| \bar{\mathbf{x}}_{k-1} - \mathbf{x}_* \right\|^2 - \left\| \bar{\mathbf{x}}_{k-1} - \bar{\mathbf{z}}_k \right\|^2 - \left\| \bar{\mathbf{x}}_k - \bar{\mathbf{z}}_k \right\|^2 + 2\eta D \frac{1}{M}\sum_{i=1}^{M}\left\| \mathbf{g}(\mathbf{z}_k^i) - \mathbf{g}(\bar{\mathbf{z}}_k) \right\| \\
&\quad + 2\left\langle \mathbf{x}_* - \bar{\mathbf{z}}_k, \frac{L}{M}\eta \sum_{i=1}^{M}\epsilon_k^i \right\rangle + 18\eta^2 \left\| \frac{1}{M}\mathbf{g}(Z_{k-1})\mathbf{1}_M - \mathbf{g}(\bar{\mathbf{z}}_{k-1}) \right\|^2 + 18\eta^2 \left\| \frac{1}{M}\mathbf{g}(Z_k)\mathbf{1}_M - \mathbf{g}(\bar{\mathbf{z}}_k) \right\|^2 \\
&\quad + 18\eta^2 L^2 \left\| \bar{\mathbf{z}}_{k-1} - \bar{\mathbf{z}}_k \right\|^2 + 6\eta^2 \|\bar{\epsilon}_k\|^2 + 6\eta^2\|\bar{\epsilon}_{k-1}\|^2
\end{aligned}
\tag{18}
$$

Noting that

$$
\left\| \bar{\mathbf{z}}_{k-1} - \bar{\mathbf{z}}_k \right\|^2 = \left\| \bar{\mathbf{z}}_{k-1} - \bar{\mathbf{x}}_{k-1} + \bar{\mathbf{x}}_{k-1} - \bar{\mathbf{z}}_k \right\|^2 \leq 2\left\| \bar{\mathbf{z}}_{k-1} - \bar{\mathbf{x}}_{k-1} \right\|^2 + 2\left\| \bar{\mathbf{x}}_{k-1} - \bar{\mathbf{z}}_k \right\|^2, \quad (19)
$$

Define $\Lambda_k = 2\left\langle \mathbf{x}_* - \bar{\mathbf{z}}_k, \frac{L}{M}\eta \sum_{i=1}^{M}\epsilon_k^i \right\rangle$. We rearrange terms in (18) and combine (19), which yield

$$
\begin{aligned}
&\left\| \bar{\mathbf{x}}_{k-1} - \bar{\mathbf{z}}_k \right\|^2 + \left\| \bar{\mathbf{x}}_k - \bar{\mathbf{z}}_k \right\|^2 - 18\eta^2 L^2\left( 2\left\| \bar{\mathbf{z}}_{k-1} - \bar{\mathbf{x}}_{k-1} \right\|^2 + 2\left\| \bar{\mathbf{x}}_{k-1} - \bar{\mathbf{z}}_k \right\|^2 \right) \\
&\leq \left\| \bar{\mathbf{x}}_{k-1} - \mathbf{x}_* \right\|^2 - \left\| \bar{\mathbf{x}}_k - \mathbf{x}_* \right\|^2 + 2\eta D \frac{1}{M}\sum_{i=1}^{M}\left\| \mathbf{g}(\mathbf{z}_k^i) - \mathbf{g}(\bar{\mathbf{z}}_k) \right\| + \Lambda_k + 6\eta^2\|\bar{\epsilon}_{k-1}\|^2 + 6\eta^2\|\bar{\epsilon}_k\|^2 \\
&\quad + 18\eta^2 \left\| \frac{1}{M}\mathbf{g}(Z_{k-1})\mathbf{1}_M - \mathbf{g}(\bar{\mathbf{z}}_{k-1}) \right\|^2 + 18\eta^2 \left\| \frac{1}{M}\mathbf{g}(Z_k)\mathbf{1}_M - \mathbf{g}(\bar{\mathbf{z}}_k) \right\|^2
\end{aligned}
\tag{20}
$$

Define $\mathbf{x}_{-1}^i = \mathbf{z}_{-1}^i = 0$ for $\forall i \in \{1, \ldots, M\}$ and $\widehat{\mathbf{g}}(\epsilon_{-1}, Z_{-1}) = \mathbf{g}(Z_{-1}) = 0_{d \times M}$. Take summation over $k = 0, \ldots, N-1$ in (20) and note that $\mathbf{z}_0^i = \mathbf{x}_0^i = 0$ for $\forall i \in \{1, \ldots, M\}$, which yield

$$
\begin{aligned}
&(1 - 36\eta^2 L^2)\sum_{k=0}^{N-1}\left\| \bar{\mathbf{x}}_{k-1} - \bar{\mathbf{z}}_k \right\|^2 + (1 - 36\eta^2 L^2)\sum_{k=0}^{N-1}\left\| \bar{\mathbf{x}}_k - \bar{\mathbf{z}}_k \right\|^2 \\
&\leq \left\| \bar{\mathbf{x}}_0 - \mathbf{x}_* \right\|^2 - \left\| \mathbf{x}_{N-1} - \mathbf{x}_* \right\|^2 + 12\eta^2\sum_{k=0}^{N-1}\|\bar{\epsilon}_k\|^2 + \sum_{k=0}^{N-1}\Lambda_k \\
&\quad + \sum_{k=0}^{N-1}2\eta D\frac{1}{M}\sum_{i=1}^{M}\left\| \mathbf{g}(\mathbf{z}_k^i) - \mathbf{g}(\bar{\mathbf{z}}_k) \right\| + \sum_{k=0}^{N-1}36\eta^2\left\| \frac{1}{M}\mathbf{g}(Z_k)\mathbf{1}_M - \mathbf{g}(\bar{\mathbf{z}}_k) \right\|^2
\end{aligned}
\tag{21}
$$

By taking $\eta \leq \frac{1}{6\sqrt{2}L}$, we have $1 - 36\eta^2 L^2 \geq \frac{1}{2}$. Take expectation and divide $N$ on both sides of (21), and then employing Lemma 1 and Lemma 3, we have

$$\mathbb{E}\left[\frac{1}{2N}\left(\sum_{k=0}^{N-1}\|\bar{\mathbf{z}}_k - \bar{\mathbf{x}}_k\|^2 + \sum_{k=0}^{N-1}\|\bar{\mathbf{x}}_{k-1} - \bar{\mathbf{z}}_k\|^2\right)\right]$$

$$\leq \frac{\|\bar{\mathbf{x}}_0 - \mathbf{x}_*\|^2}{N} + 12\eta^2 \cdot \frac{\sigma^2}{mM} + \frac{16\eta^2 DL\sigma}{\sqrt{mM}(1 - 4\eta L)} + \frac{36\eta^2\sigma^2}{\sqrt{mM}} + \frac{36\eta^2}{c-1} \cdot \frac{1}{N}\sum_{k=0}^{N-1}\mathbb{E}\|\mathbf{g}(\bar{\mathbf{z}}_k)\|^2 \tag{22}$$

By employing (17) and (22) and Lemma 1, we have

$$\frac{1}{N}\sum_{k=0}^{N-1}\eta^2\mathbb{E}\left\|\mathbf{g}\left(\frac{1}{M}Z_k\mathbf{1}_M\right)\right\|^2 \leq \frac{4}{N}\mathbb{E}\left(\sum_{k=0}^{N-1}\|\bar{\mathbf{z}}_k - \bar{\mathbf{x}}_k\|^2 + \sum_{k=0}^{N-1}\|\bar{\mathbf{x}}_{k-1} - \bar{\mathbf{z}}_k\|^2\right)$$

$$+ \frac{4\eta^2}{N}\sum_{k=0}^{N-1}\mathbb{E}\left\|\frac{1}{M}\sum_{i=1}^{M}\left(\mathbf{g}(\mathbf{z}_k^i) - \mathbf{g}(\bar{\mathbf{z}}_k)\right)\right\|^2 + \frac{4\eta^2}{NM^2}\sum_{k=0}^{N-1}\sum_{i=1}^{M}\mathbb{E}\|\epsilon_k^i\|^2$$

$$\overset{(a)}{\leq} 8\left(\frac{\|\bar{\mathbf{x}}_0 - \mathbf{x}_*\|^2}{N} + 12\eta^2 \cdot \frac{\sigma^2}{mM} + \frac{16\eta^2 DL\sigma}{\sqrt{mM}(1 - 4\eta L)} + \frac{36\eta^2\sigma^2}{\sqrt{mM}} + \frac{36\eta^2}{c-1} \cdot \frac{1}{N}\sum_{k=0}^{N-1}\mathbb{E}\|\mathbf{g}(\bar{\mathbf{z}}_k)\|^2\right)$$

$$+ \frac{4\eta^2\sigma^2}{mM} + \frac{4\eta^2}{c-1} \cdot \frac{1}{N}\sum_{k=0}^{N-1}\mathbb{E}\|\mathbf{g}(\bar{\mathbf{z}}_k)\|^2 + \frac{4\eta^2\sigma^2}{mM} \tag{23}$$

where (a) holds by (22) and Lemma 1. Divide $\eta^2$ on both sides and by basic algebras, we have

$$\left(1 - \frac{320}{c-1}\right)\frac{1}{N}\sum_{k=0}^{N-1}\mathbb{E}\|\mathbf{g}(\bar{\mathbf{z}}_k)\|^2 \leq 8\left(\frac{\|\mathbf{x}_0 - \mathbf{x}_*\|^2}{\eta^2 N} + \frac{20\sigma^2}{mM} + \frac{36\sigma^2}{\sqrt{mM}} + \frac{16DL\sigma}{\sqrt{mM}(1 - 4\eta L)}\right)$$

$$\overset{(a)}{\leq} 8\left(\frac{\|\mathbf{x}_0 - \mathbf{x}_*\|^2}{\eta^2 N} + \frac{20\sigma^2}{mM} + \frac{48(DL\sigma + \sigma^2)}{\sqrt{mM}}\right) \tag{24}$$

where (a) holds since $1 - 4\eta L \geq \frac{1}{3}$ because of $\eta \leq \frac{1}{6\sqrt{2}L}$.

Taking $c = 321$, we know that

$$\frac{1}{N}\sum_{k=0}^{N-1}\mathbb{E}\|\mathbf{g}(\bar{\mathbf{z}}_k)\|^2 \leq 8\left(\frac{\|\mathbf{x}_0 - \mathbf{x}_*\|^2}{\eta^2 N} + \frac{20\sigma^2}{mM} + \frac{48(DL\sigma + \sigma^2)}{\sqrt{mM}}\right) \tag{25}$$

$\square$

# B  Detailed Experimental Settings and More Experimental Results

**Detailed Experimental Settings**  PyTorch 1.0.0 is the underlying deep learning framework. We use the CUDA 10.1 compiler, the CUDA-aware OpenMPI 3.1.1, and g++ 4.8.5 compiler to build our communication library, which connects with PyTorch via a Python-C interface. We develop and test our systems on a cluster which has 4 servers in total. Each server is equipped with 14-core Intel Xeon E5-2680 v4 2.40GHz processor, 1TB main memory, and 4 Nvidia P100 GPUs. GPUs and CPUs are connected via PCIe Gen3 bus, which has a 16GB/s peak bandwidth in each direction. The servers are connected with 100Gbit/s Ethernet.

**More Experimental Results**  We report run-time results on CIFAR10 and ImageNet in a low-latency environment, which are presented in Figure 4. We can see that both DP-OAdam and Rand-DP-OAdam are faster than CP-OAdam.

Figure 4: Run-time comparison between OAdam, DP-OAdam, Rand-DP-OAdam and CP-OAdam for 16 learners on CIFAR10 and ImageNet in a low latency environment. The batch size used in OAdam is 256. We fix the total batch size as 256 (i.e. the product of batch size per learner and number of learners is 256). Both DP-OAdam and Rand-DP-OAdam are faster than CP-OAdam.

## C Generated Images

We compare the performance of 3 algorithms (CP-OAdam, DP-OAdam, and Rand-DP-OAdam) for training Self-Attention GAN on ImageNet by the generated images. The results are presented in Figures 5, 6, 7 respectively. All distributed algorithms (CP-OAdam, DP-OAdam, Rand-DP-OAdam) are implemented on 16 GPUs. We can see that the quality of generated images by our proposed decentralized algorithms (Rand-DP-OAdam, DP-OAdam) generate comparable images with the centralized algorithm (CP-OAdam).

Figure 5: CP-OAdam      Figure 6: DP-OAdam      Figure 7: Rand-DP-OAdam