[Reviews · NeurIPS 2020]

Review 1

Summary and Contributions: The paper discusses a decentralized stochastic optimization scheme for generic smooth min-max optimization problems with application to training GANs over large datasets, e.g. Image-net.

Strengths: Decentralized methods are becoming popular to tackle large-scale problems, and this paper studies a fairly general setup with a relatively simple algorithm, and with provable convergence guarantees. The authors tried their method on the state-of-the- art GAN approaches to well-known datasets and even compared different computational environments with high and low latency.

Weaknesses: I do not find any major issue with the paper.

Correctness: I did not closely check the proofs, but the arguments and the sketch looks convincing to me.

Clarity: It is well written.

Relation to Prior Work: Excellent citation.

Reproducibility: Yes

Additional Feedback: As I said, I do not have a major comment. The only part which confused me was the repeated local averaging for t times. Please clarify the text and also state that W^t means W to the power of t (if I am correct!) Post rebuttal comment: My opinion is unchanged about this paper.


Review 2

Summary and Contributions: ------ edit --------- 1) It is expected that in the case where M=1 exactly then the single-machine results can be recovered exactly since the analysis is a generalization of [15]. However, this seems to break down as soon as M=2, so the complexity substantially increases when going from 1 to 2 machines. This may be unavoidable but it means that even with free unlimited communications, it would be slower to use 2 machines rather than 1 with this algorithm. This seems worth commenting on, or knowing whether it's an actual feature of the algorithm or an artifact of the proof. 2) logarithmic complexity in epsilon From what I understand from the rebuttal, Corollary 1 and Remark 1 indeed only hold for communication graphs with small rho (such as the complete graph). In general the communication complexity could be much worse (\epsilon^{-8} in the case of the ring graph for example). Similarly, Corollary 1 considers multiplication by the "max degree of the network". The rebuttal seems to indicate that this is is the maximum degree of a node in case the communication matrices are stochastic but this not completely clear either and should be made more specific. If this is the case then Remark 1 would almost only hold for the random mixing strategy on the complete graph, which is far from being explicit. In particular, I think multi-consensus would actually largely degrade the result in this case since it would make the actual "max degree" equal to O(M), thus losing all hopes of logarithmic communication complexity. In any case, multi-consensus is tailored for cases when rho is close to 1. Thus, I still believe that the analysis may not be completely tight, and that Corollary 1 and Remark 1 should be substantially modified since they actually hold in a very restricted communication setting which is not specified in the paper (but which is acknowledged in the rebuttal). It would be interesting to know whether logarithmic communication is achievable in the general setting anyway if M = poly(1/epsilon). Therefore, I keep my original score. Should the paper be accepted, I believe the authors should rewrite the interpretations of Theorem 1 and be as straightforward as in the rebuttal by clearly emphasizing when the remark holds and when it does not. ------------------ This paper presents a decentralized version of the Parallel Optimistic Stochastic Gradient algorithm in which parameters are approximately averaged (in a decentralized way) before applying each gradient update. A non-asymptotic convergence theorem is given. The paper is generally well-written and includes (too?) extensive state-of-the art review. Experimental results on ImageNet and CIFAR-10 datasets are given, and seem to indicate the benefits of using a decentralized algorithm.

Strengths: Extension of the optimistic stochastic gradient algorithm to the decentralized setting, with convergence guarantees. Extensive state of the art Both low-latency and high-latency settings are investigated, for small and big models. The experiments seem rather convincing to me (though I am not an expert in GANs training).

Weaknesses: Although a convergence theorem is given, the results do not seem tight since they do not seem to recover the single-machine guarantees when M=1 (only one node). In particular, the results from Corollary 1 and Remark 1 should be compared with the results from [Theorem 1, 15]. In particular, it should be emphasized that the results from [15] are *not* recovered when M=1 (single machine setting), although the algorithm is the same. This suggests that the analysis is not tight, or otherwise should be explained more in details. The two things that prevent this recovery are that: (i) the variance term decreases as $1 / \sqrt{mM}$ instead of $1/(mM)$, and (ii) the step-size has to be chosen smaller in general even though $M=1$ (because of the $\sqrt{m}$ term). Please tell me if I have missed something and this is not the case, but this should me made clearer anyway.

Correctness: - I don't quite get the claimed $O(\log(1/\varepsilon))$ communication complexity at the busiest node. If I am correct, this is the per-iteration complexity, so the overall communication complexity has to be multiplied by N, which is of order $\varepsilon^{-8}$ in Corollary 1. - Parameter $\rho$ depends on $M$, and in general tends to $1$ as $M$ grows (except from few topologies such as the complete graph). In Remark 2 for example, it is noted that $\rho = 1 - O(1/M^2)$ for the ring graph. In this case, we roughly have that $\log(1/\rho) = O(1/M^2)$ and so it cannot be said that the convergence rate is logarithmic in $\varepsilon$ since $\log_{1/\rho}(x) = \log(x) / \log(1/\rho) = O(M \log(x))$. Have I missed something here? I believe that the derivations leading to the logarithmic communication rate from Corollary 1 and Remark 1 should be made more explicit, as I am not convinced that they hold in the current form.

Clarity: Yes it is, apart from some typos (given at the end of the additional feedback).

Relation to Prior Work: Maybe give more details about the links with extra gradient? It basically looks like extra gradient on u and v at the same time for the min max problem. This is probably worth mentioning. On a side note, I know that reference pages are not limited (which is a good thing) but more than 5 pages of references for an 8 pages paper (which is not a review paper) seems like a lot to me.

Reproducibility: Yes

Additional Feedback: - I could not find which value is used for parameter $t$ in the experiments. It seems that $t=1$ is used (which is not what theory suggests) but this is not clear. It's somewhat important because the theoretical results seem to be based on approximate but precise averaging at each step, which is far from being the case with $t=1$. - Random mixing strategies are discussed in Remark 2, but Theorem 1 is presented for fixed matrices $W$. This should be clarified. - Simply communicating with $W^t$ is generally inefficient. It is much faster to use Chebyshev acceleration (also called multi consensus) as in Scaman et al. [A], and leads to a dependence in $\sqrt{\rho}$ rather than $\rho$ for fixed topologies. It is more complicated for random topologies. - It seems that the second term in the min defining $\eta$ is always smaller than the first one, which can therefore be dropped. Similarly, since $m$ and $M$ are greater than 1, $\sqrt{mM} < 1$ so the second term in the equation is actually always smaller than the third one (and so it could be removed to improve readability, and $\sigma^2$ be replaced by $2 \sigma^2$). Overall, I think that the idea of using OSG in a decentralized fashion is great, but the results from Theorem 1 seem to be improvable in many ways. In particular, they do not seem tight since they do not recover the single-machine results when using a single machine, and the logarithmic communication complexity aspect is unclear to me. I would gladly revise my opinion if the authors clarify these 2 points in the rebuttal. Minor things, typos: Section 4: "Denote by ..." rather than "Denote ... by"? Typos: L250: an upper bound. References: [A] Scaman, K., Bach, F., Bubeck, S., Lee, Y. T., & Massoulié, L. (2017, July). Optimal Algorithms for Smooth and Strongly Convex Distributed Optimization in Networks. In International Conference on Machine Learning (pp. 3027-3036).


Review 3

Summary and Contributions: In this paper, the authors propose a decentralized parallel optimistic stochastic gradient method to train generative adversarial networks (GANs). The authors analyze its convergence, and demonstrate its applications in decentralized systems.

Strengths: To the best of my knowledge, this is the first paper that considers decentralized training of GANs. Overall, this paper is well written.

Weaknesses: 1. The convergence is in terms of \bar{z}, the average of local iterates. However, the local iterates can be quite different (namely, lack of consensus). Is there any difficulty in proving consensus of local iterates? 2. In page 7, the authors mention that “An implicit barrier is enforced at the end of each iteration so that every learner advances in a lock-step”. Please explain the meanings of barrier and lock-step.

Correctness: The proposed method and the analysis seem reasonable. The numerical experiments are convincing.

Clarity: Yes.

Relation to Prior Work: Clearly discussed.

Reproducibility: Yes

Additional Feedback: I have read the author response. My concerns have been addressed.


Review 4

Summary and Contributions: The paper proposed a gradient-based decentralized parallel algorithm for GAN traning. The algorithm seems to be the first method for distributed GAN training procedure.

Strengths: The algorithm is designed for general min-max nonconvex nonconcave problems, but is suitable for GAN.

Weaknesses: *update your review* Although the proposed algorithm seems to be the first decentralized parallel algorithm for min-max nonconvex nonconcave problems (including GAN problem), the novelty of the proposed algorithm is still not enough. The proposed algorithm and theoretical analysis are proposed, but they both are direct extensions of existing results, which significantly weaken the contribution.

Correctness: Yes

Clarity: Yes

Relation to Prior Work: In some sense.

Reproducibility: Yes

Additional Feedback:

[Author Response · NeurIPS 2020]

Thanks for all the valuable comments. Please check our responses below. We will address all minor comments.

**Reviewer 1**:

**Q1: Please clarify the text and also state that $W^t$ means $W$ to the power of t (if I am correct!).**

**A:** Thanks for the suggestion. You are absolutely correct! We will make it more clear in the revised version.

**Reviewer 2**:

**Q1: Results do not seem to recover the single-machine guarantees when M=1 (only one node).**

**A:** In the bound in Theorem 1, there are 3 terms. The only term that prevents from recovering the single-machine case is the third term (i.e., $\frac{48DL\sigma+\sigma^2}{\sqrt{mM}}$). This term completely comes from the procedure of bounding the LHS of (15) in the supplement, and is loose when $M = 1$. However, thanks to the Assumption 1(iv), when $M = 1$, the LHS of (15) is negative and disappears in the proof of Theorem 1. In terms of learning rate, when $M = 1$, we have $\rho = 0$, so $t = 0$ and hence $\eta = O(1/L)$. Hence we can exactly recover [Theorem 1, 15], if we replace $\frac{48DL\sigma+\sigma^2}{\sqrt{mM}}$ by $\frac{48DL\sigma+\sigma^2}{\sqrt{mM}}\mathbf{1}_{M>1}$ with $\mathbf{1}$ being the indicator function in the statement of Theorem 1. We will change it in the revised version.

**Q2: The claimed $O(\log(1/\varepsilon))$ communication complexity at the busiest node. If I am correct, this is the per-iteration complexity.**

**A:** You are absolutely correct. We adopt the notion 'communication complexity on the busiest node' from [5, Table 1].

**Q3: Parameter $\rho$ depends on $M$, and in general tends to $1$ as $M$ grows (except from few topologies such as the complete graph). In Remark 2, it is noted that $\rho = 1 - O(1/M^2)$ for the ring graph. In this case, we roughly have that $\log(1/\rho) = O(1/M^2)$ and so it cannot be said that the convergence rate is logarithmic in $\varepsilon$.**

**A:** Thanks for pointing it out. Indeed, for a fixed ring topology, $\rho$ is close to $1$ when $M$ is large, and in this case the per-iteration communication complexity is no longer logarithmic. However, we want to emphasize that it is indeed logarithmic in $\epsilon$ when using *the random mixing strategy with a complete graph* as in Rand-DP-OAdam (line 281–283), in which any two nodes are connected and each node randomly selects two neighbors to communicate $t$ times in each iteration. In this case, it is shown in [21] that $\mathbb{E}\|W_1 \ldots W_t - \frac{1_M}{M}\|_2 \leq \frac{\sqrt{M-1}}{(\sqrt{3})^t}$, to ensure that RHS$\leq \epsilon$, we only need $t = O(\log(1/\epsilon))$ when $M = poly(1/\epsilon)$. We make it more clear in the revised version.

**Q4: In experiments, it seems that $t = 1$ is used (which is not what theory suggests) but this is not clear.**

**A:** In theory, since $t$ is only a logarithmic term in $\epsilon$, so it is almost a constant. In practice, we set $t = 1$ and we did not incur any convergence problems. In addition, when $t = 1$, the performance already matched the centralized synchronous version of our algorithm in terms of epochs, and it has much better run-time. We will make it more clear in revision.

**Q5: Random mixing strategies are discussed in Remark 2, but Theorem 1 is presented for fixed matrices $W$.**

**A**: Thanks for the suggestion. Using the random mixing strategy does not affect the proof of Theorem 1, since the two sources of randomness (gradient noise, random mixing) can be decoupled. We will mention it in revision.

**Q6: Chebyshev acceleration (also called multi consensus) as in Scaman et al. [A].**

**A**: Thanks for pointing it out. We have already cited Scaman et al. in reference [60]. We plan to consider it for random topologies in future work.

**Reviewer 3**:

**Q1: Is there any difficulty in proving consensus of local iterates?**

**A:** We can indeed prove the $\epsilon$-consensus, since our algorithm allows $t$ rounds of decentralized communication in each iteration and $t = O(\log(1/\epsilon))$. We will mention it in revision.

**Q2: Please explain the meanings of barrier and lock-step.**

**A**: In each iteration, a learner does not proceed until it finishes exchanging and averaging its weights with its neighbors. This data dependency forms an implicit barrier (i.e., we do not need to enforce an explicit barrier in the program) so that all the learners process the same number of iterations (i.e., mini-batches) at any given time (i.e., lock-step).

**Reviewer 4**:

**Q1: The proposed algorithm and theoretical analysis are proposed, but they both are direct extensions of existing results, which significantly weaken the contribution.**

**A**: We respectfully disagree. To handle this challenging nonconvex-nonconcave min-max problem, we have to design a novel algorithm such that (1) it is simple and user-friendly to large-scale decentralized training system; (2) it can be proved to have polynomial time complexity; (3) it is able to deliver good empirical performance in large-scale GAN training. Satisfying these requirements simultaneously is difficult. **It is \*NOT\* a direct extension of any existing results. Our algorithm is carefully designed and we conduct extensive and comprehensive empirical studies.** First, we use novel algorithm design: maintaining two update sequences, designing logarithmic communication rounds, and updating the discriminator and generator simultaneously. Second, in experiments, we consider both medium-scale (WGAN-GP on CIFAR10) and large-scale (SA-GAN on ImageNet) GAN training, with both high and low latency environment, and our proposed algorithms consistently deliver remarkable performance. It would be appreciated if the reviewer can provide us concrete references so that we can compare or argue against.

[Meta-Review · NeurIPS 2020]

The paper presents a decentralized version of the Parallel Optimistic Stochastic Gradient algorithm. A non-asymptotic convergence theorem is given. The algorithm is suitable for generic smooth min-max optimization problems, including GANs. Concerns remained if the theory can recover the single-machine case, as well as more precise discussion needed about the requirements on the communication graphs (spectral gap vs max degree) and restricted communication setting. Authors confirmed in the response that the communication complexity is not logarithmic for general graphs. Also, the convergence of local iterates can be discussed better (acknowledged in the author response). We urge the authors to incorporate the feedback by all reviewers, in particular the detailed comments by Reviewer 2 for a camera ready version.